# Decoding protein phosphorylation during oocyte meiotic divisions using phosphoproteomics

Leonid Peshkin[1]*[†], Enrico maria Daldello[2†], Elizabeth S Van Itallie[1], Matthew Sonnett[1], Johannes Kreuzer[1], Wilhelm Haas[1], Marc W Kirschner[1], Catherine Jessus[2]

[1]Systems Biology Department, Harvard Medical School, Boston, United States; [2]Sorbonne Université, CNRS, Laboratoire de Biologie du Développement - Institut de Biologie Paris Seine, Paris, France

*For correspondence: Leonid_Peshkin@hms.harvard.edu

[†]These authors contributed equally to this work

Competing interest: The authors declare that no competing interests exist.

## eLife Assessment

This **important** paper describes a comprehensive quantitative phospho-proteomic analysis of the meiotic progression of *Xenopus* oocytes. Using time-resolved proteomic analyses, the authors provide insights into changes in protein levels and phosphorylation states to an unprecedented depth, quality, and quantitative detail. The key findings are **compelling** and offer a helpful resource for the scientific community.

**Abstract** Oocyte meiotic divisions represent a critical process in sexual reproduction, as a diploid non-dividing oocyte is transformed into a haploid fertilizable egg, as a prelude for the subsequent embryonic divisions and differentiation. Although cell differentiation and proliferation are governed by transcription, oocyte maturation and early embryonic divisions depend entirely on changes in protein abundance and post-translational modifications. Here, we analyze the abundance and phosphorylation of proteins during *Xenopus* oocyte meiotic maturation. We reveal significant shifts in protein stability, related to spindle assembly, DNA replication, and RNA-binding. Our analysis pinpoints broad changes in phosphorylation correlating with key cytological meiotic milestones, noteworthy changes in membrane trafficking, nuclear envelope disassembly, and modifications in microtubule dynamics. Additionally, specific phosphorylation events target regulators of protein translation, Cdk1 and the Mos/MAPK pathway, thereby providing insight into the dynamics of Cdk1 activity, as related to the meiotic cell cycle. This study sheds light on the orchestration of protein dynamics and phosphorylation events during oocyte meiotic divisions, providing a rich resource for understanding the molecular pathways orchestrating meiotic progression in the frog, and most likely applicable to other vertebrate species.

## Introduction

In mature females, the oocyte undergoes a long period of growth, during which meiosis is arrested at prophase I. In the *Xenopus* oocyte, transcription is switched off when the oocyte reaches the end of its growth phase (*Dumont, 1972*). The full-grown oocyte is transcriptionally silent and equipped with a stock of mRNAs and proteins that will support without transcription three critical embryonic processes: first, two meiotic divisions that transform the oocyte into a fertilizable haploid egg; second, the process of fertilization; and third, 12 embryonic divisions, after which transcription is initiated.

The transformation of a fully-grown oocyte arrested in prophase into a fertilizable cell, arrested in metaphase of the second meiotic division is a process called meiotic maturation. Meiotic maturation is triggered by progesterone, secreted by the follicle cells surrounding the oocyte. The secretion of progesterone is stimulated by Luteinizing Hormone produced by the pituitary. Progesterone initiates a molecular signaling cascade that lasts 3–4hr and occurs with little change in oocyte morphology. Notably, this cascade leads to the activation of the Cdk1-Cyclin B kinase, the universal inducer of cell division in eukaryotes. Cdk1 phosphorylates many substrates that promote meiotic nuclear events: nuclear envelope breakdown (NEBD), the formation of the first meiotic spindle marking the metaphase I (MI), the completion of the first meiotic division with expulsion of a polar body, and the formation of the second meiotic spindle (*Figure 1*). As in all vertebrates, the frog oocyte arrests a second time in metaphase II (MII), until fertilization stimulates the completion of the second meiotic division. The entire process of oocyte maturation occurs in the absence of transcription. The regulation of the two meiotic divisions and the 12 embryonic divisions that follow are thought to be largely regulated by changes in protein abundance and phosphorylation/dephosphorylation.

The prophase-arrested oocyte contains a store of Cdk1-Cyclin B complexes that are kept inhibited by Cdk1 phosphorylation at Y15 and T14, which are substrates of the kinase, Myt1 (*Mueller et al., 1995*). In all vertebrates, the prophase arrest is maintained by high levels of cAMP and PKA (cAMP-activated protein kinase) activity (*Figure 1*). The release of this prophase block is triggered by a drop in the levels of cAMP and the consequent inhibition of PKA, which occurs within 60min after progesterone stimulation. The identity of the critical PKA substrates whose dephosphorylation is thought to induce the pathway leading to Cdk1 activation is still unknown, with the exception of Arpp19, whose mechanism of action is not completely understood (*Dupré et al., 2014*; *Santoni et al., 2024*). Nevertheless, a drop in PKA activity leads to the synthesis of new proteins from cytoplasmic stockpiles of mRNAs as well as increases in protein level through regulation of the ubiquitination machinery (*Santoni et al., 2024*). Among the accumulated proteins are Cyclin B1, the best-known activator of Cdk1 in mitotic cells, RINGO/Speedy (*Ferby et al., 1999*; *Lenormand et al., 1999*), and Mos, a kinase specific to the oocytes (*Figure 1*). The newly synthesized B1-Cyclins bind to monomeric Cdk1 to form a small pool of active complexes that evade the inhibition of Myt1 (*Hochegger et al., 2001*; *Gaffré et al., 2011*). Subsequently, this small amount of active Cdk1 initiates a complex network of feedback loops, involving many kinases and phosphatases, thereby creating an auto-amplification loop (*Jessus, 2010*; *Figure 1*). Within this loop, the activation of the Cdc25 phosphatase that dephosphorylates Cdk1 and the inhibition of the PP2A phosphatase that counteracts Cdk1, lead to a rapid and full activation of Cdk1-Cyclin B (*Jessus, 2010*; *Lemonnier et al., 2020*). Another important player in this positive feedback loop is Mos, whose translation is induced by progesterone and which accumulates at the time of Cdk1 activation (*Sagata et al., 1989a*; *Sagata et al., 1988*; *Frank-Vaillant et al., 1999*). Mos activates Erk1/2 (also known as MAPK) indirectly, which modulates the core regulators of Cdk1 (*Haccard and Jessus, 2006*; *Figure 1*). Cdk1 and the kinases activated under its control (Mos/MAPK, Aurora-A, Plk1, etc.) trigger a second wave of protein translation and accumulation, as well as mediating the structural changes of cell division: NEBD, chromosome condensation, and formation of the MI spindle (*Jessus, 2010*; *Santoni et al., 2024*). They also activate the anaphase-promoting complex/ cyclosome (APC/C), an E3 ubiquitin ligase, leading to the degradation of both Cyclin B and securin, thus enabling entry into anaphase I, chromosome separation, and the expulsion of the first polar body (*Figure 1*). The decrease in Cdk1 activity is rapidly followed by the inhibition of APC/C, which then allows the accumulation of B-Cyclins and an increase in Cdk1 activity (*Figure 1*). These events trigger the entry into metaphase II and the formation of the MII spindle. In all vertebrates, oocytes arrest in the middle of the meiotic division as a result of the stabilization of Cyclin B, resulting from APC inhibition by the Emi2/Erp1 (*Jessus, 2010*; *Figure 1*).

The burst of phosphorylation, which occurs at NEBD when Cdk1 is activated, targets proteins belonging to three categories (*Maller et al., 1977*): first, regulators of translation and/or degradation of proteins that are essential for the progression of meiotic divisions; second, proteins directly involved in the intracellular reorganization of the oocyte, such as the components of meiotic spindles, the nuclear envelope or other cytoskeletal elements; third, proteins that do not have a role in meiotic divisions but regulate fertilization and early embryonic development. Notably, the sperm does not contribute any proteins or RNA except for its genome and its two centrioles. With fertilization, there is extrusion of the second polar body, the fusion of the male and female pronuclei. Within a few

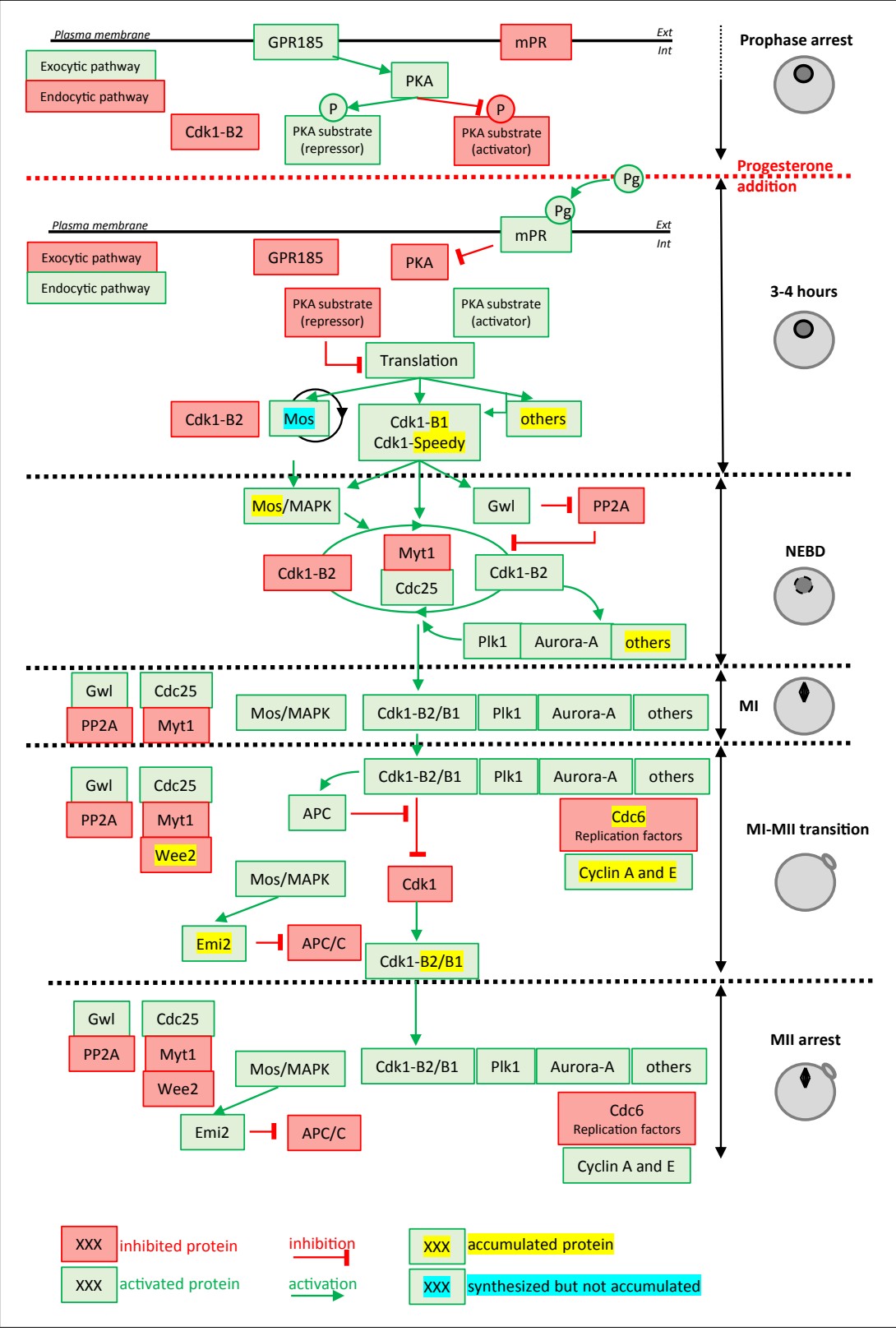

**Figure 1.** Molecular events coordinating *Xenopus laevis* oocyte meiotic divisions. From top to bottom: The prophase arrest is maintained by high PKA activity due to the constitutively expressed GPR185 receptor. The exocytic pathway is active while the endocytic pathway is downregulated. Cdk1-Cyclin B2 complexes are present under an inactive state due to Cdk1 phosphorylation. Progesterone activates the mPRβ receptor, GPR185 is inactivated, the exocytosis and endocytosis pathways are inverted. PKA is inactivated and promotes translation and/or accumulation of new proteins, among them

*Figure 1 continued on next page*

*Figure 1 continued*

Mos (that does not accumulate due to its instability) and Cyclin B1 (due to its stabilization). New Cyclin B1 proteins associate with Cdk1 and form a starter amount of Cdk1 activity. This Cdk1 starter stabilizes Mos and promotes Cdk1-Cyclin B2 activation through the regulation of Myt1, Cdc25, Gwl, and PP2A. The autoamplification loop is launched, whereby more Cdk1 is activated, more it activates kinases as Mos-MAPK, Aurora-A, and Plk1, all of them contributing with Cdk1 to regulate Myt1 and Cdc25, hence accelerating Cdk1 activation. Phosphorylated substrates trigger nuclear envelope breakdown (NEBD) and the metaphase I (MI) spindle assembly as well as protein translation. Anaphase-promoting complex (APC) is activated under the control of Cdk1 and leads to Cyclin B degradation. The Cdk1 activity decreases and allows anaphase I and expulsion of the first polar body. Under the Mos/MAPK control, Emi2/Erp1 inactivates APC, halting Cyclin B degradation. The re-accumulation of Cyclin B increases Cdk1 activity that triggers the formation of the metaphase II (MII) spindle, avoiding the assembly of a nucleus and DNA replication. APC remains inhibited by Emi2/Erp1, what ensures the oocyte arrest in MII. Color code: red: inactivated proteins; green: activated proteins; red arrow: inhibition; green arrow: activation; yellow highlight: accumulated protein; blue highlight: protein synthesized but not accumulated.

minutes after fertilization, the first S-phase starts, followed by the first 12 embryonic division cycles, all occurring in the absence of transcription (*Blitz and Cho, 2021*). Thus, these events depend exclusively on maternal proteins and mRNA. Phosphoregulation is a central and conserved mechanism that enables remodeling of the oocyte proteome and supports cell division. It prepares the oocyte to transform into the egg and undertake development. In the absence of transcription, our knowledge of the dynamic patterns of phosphorylation of thousands of oocyte phosphosites represents an essential tool for the study of oogenesis and embryogenesis, especially the unique meiotic and mitotic processes. Interestingly, two recent studies based on phosphoproteomics have revealed new insights into how phosphorylation dynamics regulate meiotic divisions in yeast (*Celebic et al., 2024*; *Koch et al., 2024*) and mouse (*Sun et al., 2024*). We now extend this analysis to the physiological process of oogenesis specific to non-mammalian metazoans. The study of mitotic processes in somatic cells, which are short in duration, requires exquisite synchronization, which is difficult. Although previous studies of protein phosphorylation during mitosis have revealed the identity of numerous players of cell division, especially those controlling the structural reorganization of the dividing cell and its checkpoints, they have not been able to fully elucidate the epistatic relationships between kinases and phosphatases, which are nonetheless crucial to understanding the temporal sequences regulating the various events of mitosis. This weakness is due to the fact that cell proliferation is a continuous phenomenon, without physiological arrest. Its duration varies from one cell to another within the same population. The necessity of using synchronizing agents induces non-physiological arrests, which activate checkpoints, leading to the resumption of the cell cycle not comparable to that of untreated cells. Moreover, these studies often use transformed cell lines with accumulation of mutations that makes it hard to compare with cells in a physiological cellular context. The exceptional features of oocyte maturation for experimental studies of mitosis and meiosis are: (1) the synchronicity of the process in the oocyte, initiated by a physiological signal, progesterone, (2) the natural oocyte arrests in oocyte maturation at either prophase I (equivalent to a late G2-arrest) or at MII (M-phase arrest), (3) and the large size of the oocyte (1.2mm in diameter) and its high protein content (30µg soluble proteins per cell), offering the rare opportunity of single-cell proteomics. In addition, the *Xenopus* model offers great assets. From a technical perspective, in the mouse, the small number of oocytes and their low protein content limit proteomic approaches (30,000 oocytes isolated from 950 mice were required for the phosphoproteomic profiling by *Sun et al., 2024*). Furthermore, the resumption of meiotic divisions in

mouse obeys a particular regulation: unlike other mammalian models (including the human species) and other vertebrates whose entry into meiosis I depends strictly on protein translation, it does not require the synthesis of new proteins, making it a somewhat marginal model (*Meneau et al., 2020*). For these reasons, we performed time-resolved proteomics and phosphoproteomics in *Xenopus* oocytes from prophase I through the MII arrest of the unfertilized egg.

In this study, we have analyzed the changes in the phosphorylation in relation to three categories of events: specific periods of meiotic maturation, activities of the master regulators of meiosis, and

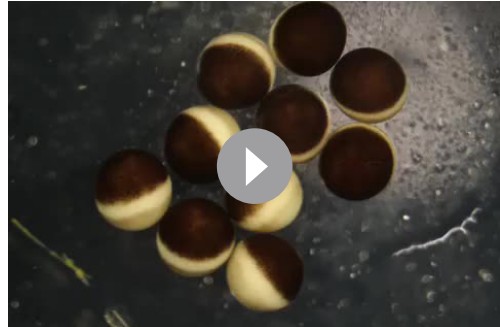

**Video 1.** Meiotic maturation of *Xenopus* oocyte.
https://elifesciences.org/articles/104255/figures#video1

peculiar cellular features of meiosis. Our analysis confirms the high quality of our phosphoproteome by the detection of multiple phosphorylation events with previously documented functional effects on meiotic divisions or early development. Through the detection of many new regulated phosphosites, it provides a rich pool of candidate proteins for multiple avenues of investigation of previously uncharacterized important players of the oocyte to embryo transition.

## Results

### The temporal resolution of the events of meiotic divisions

To determine both proteome and phosphoproteome of *Xenopus* oocytes during meiosis resumption, meiotic maturation was triggered by progesterone, and oocytes were collected at regular intervals after progesterone exposure (*Video 1*, Methods): prophase oocytes (PRO) at the time of progesterone addition, then 2hr after progesterone treatment, with hourly timepoints past that, up to 9hr post-treatment. Groups of ten or fifty oocytes were collected at each timepoint for protein and phospho-proteomics measurements, respectively. After oocyte lysis, digestion into peptides and phospho-peptide enrichment in the case of phosphoproteomics analysis, samples were processed for TMT-MS3 LC/MS. After isolating phosphorylated proteins, they were chemically tagged with barcoding labels (TMT) to run multiple samples through the instrument simultaneously, thus minimizing artifacts. The resulting merged sample is then analyzed using liquid chromatography combined with a mass spectrometry technique (MS3) to accurately measure relative phosphorylation levels across the barcoded samples (Methods).

To correlate the time points of this experiment with the cytological events known to occur during meiosis resumption, a set of proteins whose accumulation had been extensively studied during *Xenopus* oocyte meiosis progression were used as markers. One of the first events following progesterone stimulation is the accumulation of Cyclin B1 that begins before and independently of Cdk1 activation (*Santoni et al., 2024*; *Frank-Vaillant et al., 1999*). In our experiments, Cyclin B1 accumulation starts at about 2hr and increases linearly until MII arrest at 6.5hr (*Figure 2A*). It has been reported that Mos translation is stimulated by progesterone concomitant with the accumulation of Cyclin B1; however, the Mos protein does not accumulate before NEBD because it turns over rapidly (*Santoni et al., 2024*; *Frank-Vaillant et al., 1999*). Another critical well-documented event that occurs downstream to Cdk1 activation is the degradation of CPEB1 (*Santoni et al., 2024*; *Mendez et al., 2002*), an RNA-binding protein whose degradation activates protein translation in MI. Hence, Mos accumulation and the degradation of CPEB1 are well-established markers for the timing of NEBD. After NEBD, activation of the anaphase-promoting complex/cyclosome (APC/C) leads to the ubiquitination and degradation of Cyclin B2, which marks the entry into anaphase I. During the MI-MII transition, multiple additional events take place: Cyclin B3 is degraded (*Bouftas et al., 2022*), Cdc6 (*Lemaître et al., 2002*; *Whitmire et al., 2002*; *Daldello et al., 2015*), Wee2 (*Nakajo et al., 2000*), and Cyclin E (*Rempel et al., 1995*) accumulate, reaching their maximal levels in MII. These four events mark the entry into MII. Based on these markers, the timing of cytological events in our experiment is as follows (*Figure 2A*): NEBD and MI occur between 3 and 4hr, the MI-MII transition takes place between 5 and 6hr, and entry and arrest in MII are at 7hr. This time course is in agreement with the cytological previously described events (*Huchon et al., 1981*; *Gard, 1992*). A physiological measure of MII arrest can be ascertained by the ability of the oocytes to be activated by an electric shock, a property that is acquired only in MII (*Video 2*).

### Validation of the quantitative proteomic data

The quality of oocytes resuming meiosis is also confirmed by the dynamics of protein accumulation and degradation that occur during meiotic progression (*Figure 2B*). A set of 34 proteins, whose changes in concentration have been previously studied during meiotic divisions, was used to validate the quality of our quantitative proteomics data (*Figure 2B—figure supplement 1A*). Our proteome correctly classifies 15 out of 18 proteins accumulating during *Xenopus* meiosis resumption (FC(MII/PRO)>1.5), 11 out of 12 proteins whose concentration is constant (0.75<FC(MII/PRO)<1.5), and 4 out of 4 proteins whose accumulation was reported to decrease (FC(MII/PRO)<0.75) (*Figure 2B—figure supplement 1A*). The concordance between our dataset (30/34 proteins) with the changes in the protein levels reported in the literature is substantially higher than in a previously published

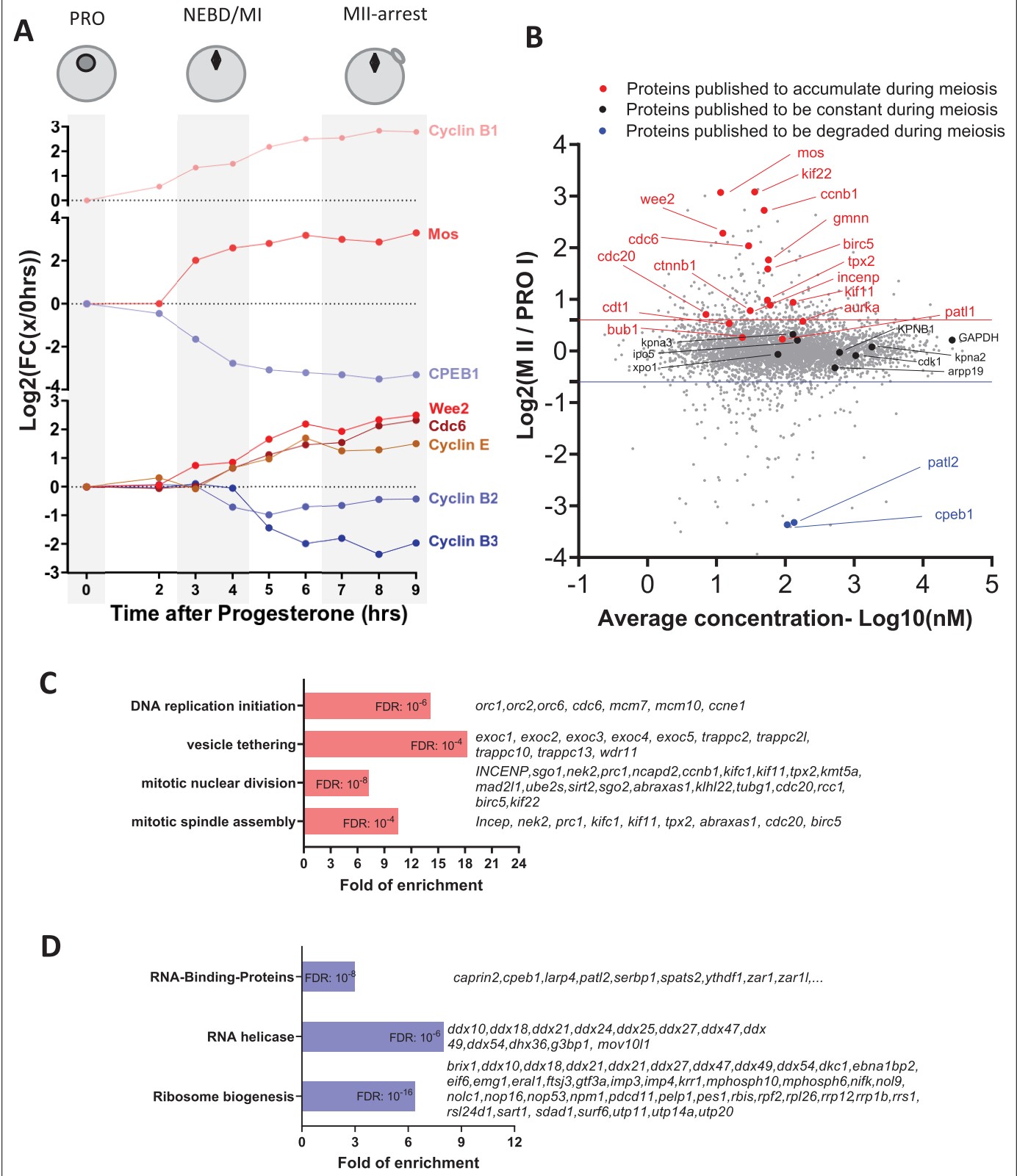

**Figure 2.** Changes in the protein levels during meiotic divisions. (**A**) The relative changes in the concentration of proteins whose accumulation/degradation has been reported to correlate with the critical stages of *Xenopus* meiotic divisions have been plotted. (**B**) Changes of protein stoichiometry during meiotic maturation. The log2 fold changes between the concentration of each protein in prophase (PRO)-arrested oocytes and metaphase II (MII)-arrested oocytes were plotted against the average concentration during meiotic divisions. The red line marks the 1.5-fold change

*Figure 2 continued on next page*

*Figure 2 continued*

used as a threshold to identify accumulating proteins. The blue line marks the 0.75-fold change used as a threshold to distinguish proteins that are degraded during meiosis. Proteins, whose changes in homeostasis during meiotic divisions are documented in the literature, are marked with the following color code: red, for proteins accumulating; black, for proteins which are expressed at a stable level; and blue, for proteins that are degraded. (C–D) Gene ontology analysis of the biological processes enriched among the proteins found to accumulate (in red, panel C) or degrade (in blue, panel D) during meiosis. The statistical significance of the enrichment is expressed as false discovery rate (FDR). A representative subset of the proteins belonging to each group is displayed.

The online version of this article includes the following figure supplement(s) for figure 2:

**Figure supplement 1.** Validation of the proteomic dataset.

**Figure supplement 2.** Distribution of absolute protein abundance in the oocyte by functional set.

**Figure supplement 3.** Clusters of dynamics of oocyte protein expression and phosphorylation in response to progesterone.

**Figure supplement 4.** Protein degradation during meiotic maturation: ribosomal proteins, E3 ligases targets.

system-wide proteomic dataset from *Xenopus* oocytes where only 7 out of these 34 proteins were correctly measured (*Peuchen et al., 2017*; *Figure 2—figure supplement 1A–B*). This also may explain why changes in protein concentration reported in that study poorly correlate with the one measured in our proteomic study (*Figure 2—figure supplement 1B*). The high quality of the new proteomic data is also confirmed by the strong correlation of the translational pattern of *Xenopus* allo-alleles (*Figure 2—figure supplement 1C–D*), which were derived by genome endoduplication and are called 'S' or 'L' alleles (*Session et al., 2016*).

## Protein homeostasis during meiotic divisions

### Absolute concentrations of proteins during meiotic maturation

The absolute concentration of 7974 proteins identified in our proteome dataset was calculated (See Appendix 1 and *Supplementary file 1*). We then performed a qualitative comparison of average oocyte abundance of proteins belonging to distinct functional groups (*Figure 2—figure supplement 2*). As expected, the less abundant subgroups include signaling molecules and transcription factors. Perhaps of more surprise, E3 enzymes and kinases are not abundant. At the other end of the spectrum are glycolytic and tricarboxylic acid cycle enzymes, as well as proteasome and ribosome components. These observations are expected because the oocyte has a stock of nutrient molecules and all the enzymes linked to their metabolism, which are used in viviparous animals throughout embryonic and larval development. For instance, the glycolytic enzyme GAPDH that catalyzes an important energy-yielding step in carbohydrate metabolism, is a very abundant protein in the oocyte (average concentration of between 8.88µM and 17.83µM). The ribogenesis program, which takes place, before meiosis resumption, during the oocyte growth from stage III to stage VI, according to Dumont classification (*Dumont, 1972*), allows the accumulation of very large quantities of ribosomal RNAs and proteins that are used during embryonic development, until the swimming tadpole stage (*Woodland, 1974*). Similarly, the components of the DNA replication machinery are quite abundant, probably in anticipation of the post-fertilization cell cycles. All these data are, therefore, consistent with the biological understanding of their roles during embryonic development after post-fertilization (*Peshkin et al., 2015*).

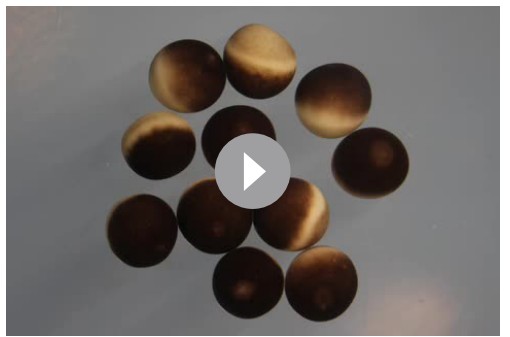

**Video 2.** Activation of *Xenopus* metaphase II oocytes. https://elifesciences.org/articles/104255/figures#video2

### Temporal profiles of protein levels

We clustered the relative data for protein abundance across subsequent hourly timepoints post-progesterone stimulation (see Methods). As illustrated in *Figure 2—figure supplement 3*, the majority of the 60 expression profiles are flat, demonstrating that most (over 80%) proteins are stable during meiotic maturation. Our proteomic dataset allowed to identify 12% of proteins affected by changes in concentration during meiotic progression: 413 proteins that accumulate and 562 proteins whose concentration decreases (*Figure 2B—figure supplement*

*1E*; see *Supplementary file 2*). Although these changes only concern a minority of proteins, they are critical for the meiotic process. Since transcription is silent during this period, these documented changes must result from the regulation of translation or/and degradation (*Meneau et al., 2020*).

## Protein accumulation: the machinery of cell division and DNA replication

Gene ontology analysis was used to characterize the functional significance of the 413 proteins whose level increases during meiosis. Proteins involved in spindle assembly and mitotic division were significantly enriched (*Figure 2C*). Among these important regulators are the Targeting-protein-for-Xklp2 (Tpx2) and Protein-regulator-of-cytokinesis 1 (Prc1) that also accumulate during meiotic maturation in mouse oocytes (*Brunet et al., 2008*; *Li et al., 2021*). We also find that the centrosomal maturation factor, SSX2IP, a plus-end motor kinesin, Kif11/Eg5, and a minus-end one, Kifc1 (*Houliston et al., 1994*), also accumulate significantly during meiotic maturation. Three of these proteins, Tpx2 and the two kinesins, play an important role in the assembly of both acentrosomal meiotic spindles and centrosomal mitotic spindles (*Gruss, 2018*; *Kufer et al., 2002*; *Ems-McClung et al., 2020*; *Miller et al., 2019*). Interestingly, the prophase oocyte is equipped with most of the proteins essential for the formation of division spindles. Hence, these new results suggest that increases in the relative level of some components are required to support the meiotic and first rapid embryonic divisions.

Another protein category found to increase is the machinery of DNA replication initiation (*Figure 2C*). Indeed, oocytes accumulate the components of the DNA replication machinery during meiosis resumption to support the 12 rapid rounds of cell cycles occurring during early embryogenesis, all of which occur in the absence of transcription. In *Xenopus*, Cdc6 is a component of the pre-replicative (pre-RC) complex that is not detectable in the prophase-arrested oocyte by western blot (*Lemaître et al., 2002*; *Whitmire et al., 2002*; *Daldello et al., 2015*). Cdc6 accumulation occurs during the MI-MII transition to confer oocytes the competence to replicate DNA (*Lemaître et al., 2002*; *Whitmire et al., 2002*; *Daldello et al., 2015*; *Figure 1*). Translational and then post-translational mechanisms targeting Cdc6 prevent the pre-RC from functioning between the two meiotic divisions, thus ensuring the production of a haploid gamete. Our proteomic analysis shows that additional pre-RC proteins, such as ORC1, ORC2, and ORC6 accumulate during meiotic maturation, revealing a strong co-regulation of the components of the pre-RC.

Another process controlled by highly enriched proteins is vesicle tethering, which includes the family of exocyst proteins (*Figure 2C*). Interestingly, exocyst proteins are involved in the completion of cell division during the secretory-vesicle-mediated abscission (*Gromley et al., 2005*). The accumulation of these proteins during meiotic maturation could be important to support the extrusion of the polar bodies, as already suggested in *C. elegans* (*Kumar et al., 2019*) and the rapid cell divisions sustaining early embryogenesis.

## Protein degradation: RNA-binding proteins (RBPs)

Gene ontology analysis of the 562 proteins whose concentration decreases during meiosis reveals that 116 (21%) are annotated to bind RNA. Among them, a large portion is RNA helicases (*Figure 2D*). Among these RBPs, another important group are proteins that are involved in ribosome biogenesis. The ribogenesis program takes place during the oocyte growth, allowing the accumulation of ribosomal RNAs and proteins that are used during embryonic development (*Woodland, 1974*). This process is downregulated during meiotic maturation (*Hyman and Wormington, 1988*; *Chen et al., 2011*). The decrease in the concentration of proteins involved in rRNA synthesis and processing does not result in a general decrease of ribosomal proteins that are stable during meiotic divisions (*Figure 2—figure supplement 4A*). Halting ribogenesis may allow the oocyte to re-allocate its energy resources to the production of other cellular components required for meiotic divisions and embryogenesis. Intriguingly, Rpl26 (Ribosomal protein L26) is the only ribosomal protein detected to decrease during meiotic divisions (*Figure 2—figure supplement 4A*). Since this protein is known to be dispensable for ribosome formation in yeast (*Babiano et al., 2012*; *Warner and McIntosh, 2009*; *Takagi et al., 2005*), the distinct regulation of Rpl26 as compared to the other Rpls suggests that Rpl26 might have extra-ribosomal functions. This hypothesis is reinforced by its specific modification by the ubiquitin-like UFMylation modification (*Walczak et al., 2019*) and its role in the regulation of protein stability (*Zhang et al., 2016*). Another subgroup of RBPs that decrease during meiotic divisions is involved in the repression of translation during prophase arrest, such as CPEB1 (*Mendez et al., 2002*;

*Reverte et al., 2001*; *Piqué et al., 2008*), PATL2 (*Nakamura et al., 2010*; *Cao et al., 2021*; *Christou-Kent et al., 2018*), and Zar1l (Zar2) (*Charlesworth et al., 2012*; *Yamamoto et al., 2013*; *Rong et al., 2019*). Remodeling the repertoire of RBPs in the oocyte could be a general mechanism to regulate translation during meiotic divisions. Among these RBPs is Serbp1, which in mouse is involved in the stabilization of mRNA involved in meiosis resumption (*Chew et al., 2013*) and in ribosome hibernation (*Leesch et al., 2023*), as well as Caprin2, whose degradation is also observed during meiotic maturation of human oocytes (*Virant-Klun et al., 2016*). Caprin2 is a RBP with unknown function and highly enriched in the oocyte Balbiani body (*Boke et al., 2016*), a non-membrane compartment specific to the early diplotene oocytes, which contains mitochondria, RNAs, and endoplasmic reticulum. The functions of this organelle are not clear, although it is strongly correlated in *Xenopus* to the establishment of oocyte polarity and to the localization of maternal determinants (*Jamieson-Lucy and Mullins, 2019*). In *Xenopus*, it disperses at the onset of vitellogenesis, leaving a wedge-shaped cytoplasmic region rich in mitochondria (*Wilk et al., 2005*) and promotes the formation of RNP condensates in the vegetal cortex of the oocyte (*Yang et al., 2022*). Interestingly, Rbpms2, another regulator of the Balbiani body (*Kaufman et al., 2018*), is strongly degraded during meiotic maturation, highlighting how dynamic during meiosis is the composition of this transient compartment specific to the oocyte.

Interestingly, distinct E3 ubiquitin ligases, mainly APC and SCF (Skp, Cullin, F-box containing complex) have been shown to be active at different time periods of meiosis resumption in *Xenopus* oocytes (*Kinterová et al., 2022*). 186 proteins (37%) identified as decreasing in our proteome include in their sequences one of the APC degrons (*Figure 2—figure supplement 4B*). This group of proteins includes two known APC targets, Cyclin B2 (*Glotzer et al., 1991*) and Securin (*Cohen-Fix et al., 1996*). Additionally, 30 proteins (6%) whose concentration decreases in the oocytes have one SCF degron in their sequence (*Figure 2—figure supplement 4B*), including CPEB1 (*Reverte et al., 2001*) and Cdc6 (*Daldello et al., 2015*), two characterized SCF-targets. Interestingly, 253 proteins (57%) of proteins that decrease during meiotic divisions do not bear in their sequence any of the degrons for the ubiquitination systems identified in *Xenopus* oocytes (*Figure 2—figure supplement 4B*). Further research is, therefore, required to identify the sequences and the E3 enzymes and deubiquitinating enzymes that control the turnover of this important protein class.

## The highly dynamic landscape of phosphorylation during meiotic maturation

We performed phosphoproteomic mass spectrometry analysis in order to measure the overall dynamics of protein phosphorylation taking place during meiotic divisions and to identify which specific proteins and which sites are phosphorylated during this process. We identified 6783 different phospho-peptides and quantified the dynamics of relative change across all time points (*Supplementary file 3*). These peptides came from 2308 distinct proteins. Our phosphosites dataset contains 80% Ser, 19.9% Thr, and 0.01% Tyr. Phospho-Tyr is slightly less abundant than what has been described in most cells (up to 0.05% *Sharma et al., 2014*). The same observation was made regarding the distribution of phosphorylated amino acids in mouse oocytes, where phospho-Tyr abundance is relatively diminished in oocytes compared to mouse organs (*Sun et al., 2024*). When respective protein and unphosphorylated peptide were also measured, we were able to compute the stoichiometry (see Methods). A list of all 415 phosphosites for which it was possible to calculate the phospho-occupancy is provided (*Supplementary file 4*).

We identified the human homologous counterparts of 5901 (87%) among the 6783 sites found to be phosphorylated during meiotic divisions. Interestingly, only 177 (3%) out of 5901 conserved sites were already known to be phosphorylated on phosphosite.org database (https://www.phosphosite.org/homeAction). Thus, our phosphosite dataset identifies a very large number of novel phosphosites, never previously identified and characterized. Such a small number of previously reported phosphorylations among those detected by our phosphoproteome might suggest that a large portion of these phosphorylation events are related to cell division. Dividing cells generally account for a small fraction of tissue mass. This leads to a high dilution of the phosphorylation signature of cell division, as compared to the highly synchronous meiotic divisions of oocyte maturation. Interestingly, 60% of the phosphosites detected in oocytes are dynamically regulated during meiotic maturation, highlighting the importance of this post-translational regulation in controlling this last step of oogenesis and meiosis. Unbiased clustering reveals that changes in protein phosphorylation are pervasive and far

more dynamic than changes in protein abundance (*Figure 2—figure supplement 3*), which was also noted during yeast meiosis (*Koch et al., 2024*). Comparing the three stages of meiotic division (PRO, MI and MII) and based on the main cellular events of these stages, we can group phospho-peptide stoichiometry into 5 classes (*Figure 3A*; *Supplementary file 3*). Class I includes proteins bearing a phosphosite whose occupancy decreases during the first 2hr following progesterone stimulation. This class includes only 136 phospho-peptides (2%) which might be involved in the early signaling pathway induced by progesterone and PKA downregulation. This small number was expected since there is a low level of phosphorylation detectable in prophase-arrested oocytes (*Maller et al., 1977*). This makes sense since there are presumably only a few substrates of PKA sufficient to initiate meiosis resumption. Class II is larger (39% of the phospho-peptides) and includes proteins whose phosphorylation increases between NEBD and MI and which remain highly phosphorylated in both MI and MII. The massive amount of Class II phosphosites presumably reflects the catalytic power of Cdk1 as well as its downstream kinases, to generate the thousands of substrates essential for cell division. We used the online server (https://kinase-library.phosphosite.org/kinase-library/score-site) to score the phosphosites in the Class II and predict the kinases likely responsible for their phosphorylation (*Johnson et al., 2023*). Using a percentile score threshold of 90, we identified 303, 304, and 267 peptides predicted to be phosphorylated by Cdk1, Erk1/2, and Plk1, respectively (*Figure 3—figure supplement 1A*). Among these, 166 sites were predicted to be efficiently phosphorylated by either Cdk1 or Erk1/2, consistent with the similarity in their phosphorylation consensus motifs. In contrast, there was minimal overlap between the sites predicted to be phosphorylated by Plk1 and those targeted by Cdk1 or Erk1/2. Consequently, Class II phosphorylations are largely dependent on Cdk1 activity, whether they are directly catalyzed by this kinase or indirectly through kinases under its control, such as Erk1/2 and Plk1 (see below). Our measurements show that most of the sites, which are phosphorylated in MI, remain stably phosphorylated during MI to MII transition is consistent with the observations made in yeast showing that most of Cdk1 motifs remain phosphorylated at the end of meiosis I (*Celebic et al., 2024*). Class III (5%) consists of proteins whose phosphorylation peaks in MI then decreases while oocytes progress throughout meiotic maturation. This is the most predominant class of phosphosites present during meiotic maturation in starfish (*Swartz et al., 2021*) and follows the peculiar pattern of Cdk1 activity peaking at MI in these oocytes (*Okano-Uchida et al., 1998*; *Kishimoto, 2003*). Class IV (9%) is composed of proteins whose phosphorylation progressively increases from MI to MII, hence potentially implicated in MII entry. Finally, Class V (5%) comprises proteins whose phosphorylation increases from MII and which could play a role in establishing the MII arrest. The phosphosites that are specific for either MI (Class III) or MII (Classes IV and V) are much less abundant (5%, 9%, and 5%, respectively, *Figure 3A*) and might distinguish the biochemical state of the two meiotic divisions. These conclusions were supported by the identification of the kinases whose predicted phosphorylation sites are most enriched within each class, by using the same server (https://kinase-library.phosphosite.org/kinase-library/score-site). Each class was associated with a distinct set of enriched kinases (*Figure 3—figure supplement 1B*). It confirms that Cdk1 and its downstream kinases, Erk1/2 and Plk1 are the master regulators in Class II. The kinases identified in Classes IV and V may represent novel regulators of entry into and arrest at MII.

## The release of the prophase arrest: receptors and membrane trafficking
### Receptors

Since meiotic maturation signaling begins with the interaction of progesterone with membrane receptors, we first focused our analysis on these proteins (*Josefsberg Ben-Yehoshua et al., 2007*; *Sadler and Maller, 1985*). Progesterone signals through activation of a seven-pass-transmembrane progesterone receptor (mPRβ or PAQR8) that belongs to the progestin and adiponectin receptor family (*Josefsberg Ben-Yehoshua et al., 2007*; *Nader et al., 2018*; *Nader et al., 2020*; *Figure 1*). The release of the prophase block also involves the inactivation of an orphan constitutively active GPCR, GPR185, responsible for maintaining oocyte prophase arrest by ensuring high cAMP levels and PKA activity (*Ríos-Cardona et al., 2008*; *Deng et al., 2008*). Progesterone decreases GPR185 signaling either through its cleavage by a metalloproteinase (*Deng et al., 2008*) or through its endocytosis (*Nader et al., 2014*). One limitation of our dataset lies in the loss of some membrane proteins during the fractionation protocol, explaining why our proteomic workflow did not detect mPRβ and GPR185 or its close relative GPR12. Nevertheless, it does detect some proteins that interact with mPR, i.e.,

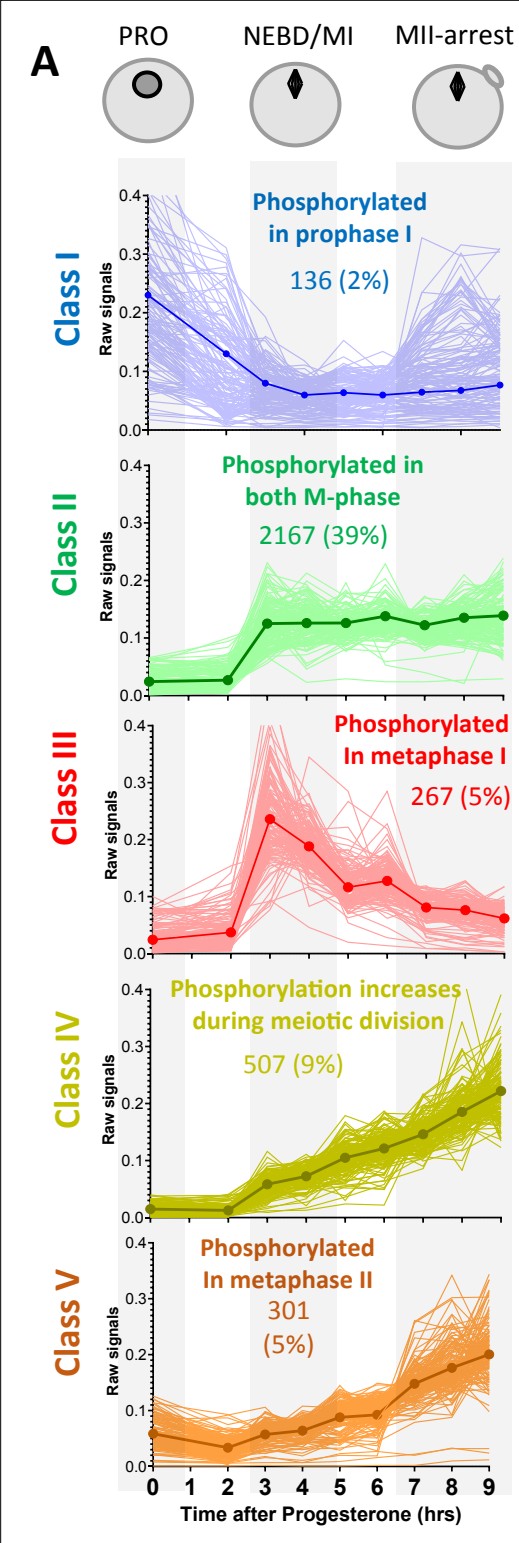

## Class I proteins

| | gene name | description | PMID |
|---|---|---|---|
| Vessicle trafficking and membrane organization | C2CD2L | an endoplasmic reticulum protein that functions both as membrane tether to the plasma membrane and as phospholipid transporter | 32693056 |
| | MAGI1 | a scaffolding protein localized at the plasma membrane | 35389514 |
| | NKTR | a membrane-anchored protein with function in protein folding | 7906541 |
| | LDLRAP1 | interacts with the cytoplasmic tail of the LDL receptor and facilitates the clathrin-mediated endocytosis of LDLR and other members of this family | 15728179 |
| | PI4KB | PKA substrate that regulates Golgi reorganization during mitosis, and is involved in Golgi-to-plasma membrane trafficking | 23712958 |
| Actin cytoskeleton | WIPF2 | induces reorganization of the actin filament network around the plasma membrane by cooperating with WASP/WASL88 | 21785420 |
| | SPTBN1 | an actin crosslinking and molecular scaffold protein that is involved in Ca2+ and calmodulin-dependent exocytosis | 32601107 |
| | ITPRID2 | tethers IP3 receptors to the actin under the plasma membrane to mediate Ca2+ signalling | 34301929 |
| Translation | PATL1 | RNA binding protein involved in mRNA stabilization and translation repression | 20826699 |
| | eIF4ENIF1 | binds and sequesters eIF4E inhibiting protein translation initiation. | 16157702 |
| | Maskin | component of the translation repression complex nucleated by CPEB1 that is involved in mantaining translation repression in Prophase arrested oocytes. | 10635326 |
| Ubiquitin ligases | Ube2O | E2/E3 ubiquitin-protein ligase with both E2 and E3 ligase activities. | 30468556 |
| | Faf2 | endoplasmic reticulum–associated protein involved in the dissolution of RNA stress granules | 34739333 |
| | Ube4b | E3-ubiquitin ligase, with unknow function. | - |
| | DCUN1D5 | catalyzes the neddylation of cullins that is required for the activation of the cullin-E3 ubiquitin ligase. | 26906416 |

**Figure 3.** Changes in protein phosphorylation during meiotic divisions. (**A**) Relative phosphopeptide signals were normalized for the changes in the total level of the proteins. The changes in the phosphopeptides were measured between the three biological stages of meiotic divisions (PRO = 0hr, NEBD/MI = average of 3 and 4hr, and MII-arrest=average of 7–9hr). Class I (n=136–2%, in blue) includes proteins bearing a phosphosite whose occupancy decreases during meiosis resumption (Log2FC(MI/PRO)<-1). Class II (n=2167–39%, in green) contains proteins that are highly phosphorylated from MI (Log2FC(MI/PRO)>1) to MII (–1<Log2FC(MII/MI)<1). Class III (n=267–5%, in red) consists of proteins whose phosphorylation is specific to MI

*Figure 3 continued on next page*

*Figure 3 continued*

(Log2FC(MI/PRO)>1, Log2FC(MII/MI)<-1). Class IV (n=507–9%, yellow) is composed of proteins whose phosphorylation increases from MI (Log2FC(MI/PRO)>1) to MII, peaking in MII (Log2FC(MII/MI)>1). Class V (n=301–5%, orange) comprises proteins whose phosphorylation does not vary in MI (–1<Log2FC(MI/PRO)<1) but are phosphorylated only in MII (Log2FC(MII/MI)>1). The average signal of each Class was plotted with a thick and dark line. (**B**) Table summarizing the proteins bearing phosphosites belonging to Class I.

The online version of this article includes the following figure supplement(s) for figure 3:

**Figure supplement 1.** Predicted kinases from the phosphorylation sites.

PGRMC1, VLDLR and APPL1. PGRMC1 is known to form a receptor complex with mPRα and is necessary for mediating progesterone signaling in zebrafish oocytes (**Wu et al., 2018**). Interestingly, our phosphorylation data reveals that PGRMC1 and its close relative PGRMC2 are phosphorylated on two homologous threonine residues during meiotic maturation, suggesting that these proteins may contribute to mPRβ activity in *Xenopus* oocytes. In the prophase-arrested oocyte, VLDLR regulates mPR trafficking from the endoplasmic reticulum through the Golgi to the plasma membrane, preparing the oocyte to be responsive to progesterone (**Nader et al., 2018**). In contrast, progesterone induces clathrin-dependent endocytosis of mPRβ into signaling endosomes, where mPR interacts transiently with APPL1 and Akt2 to induce meiosis (**Nader et al., 2020**). The phosphoproteomic data reveal that both APPL1 and Akt are phosphorylated in response to progesterone but at the time of Cdk1 activation and on residues distinct from those identified by **Nader et al., 2020**. Hence, phosphorylation events may continue to regulate the trafficking and activity of this receptor during meiotic maturation, ensuring that the plasma membrane of the future egg is devoid of any steroid receptor.

## Membrane traffic

Vesicular trafficking at the cell membrane appears to be crucial for the maintenance and the release of the prophase meiotic arrest, and may also function through early embryogenesis (**Figure 1**). In the prophase-arrested oocyte, the accumulation of the GPR185 receptor at the plasma membrane maintains high cAMP levels (**Nader et al., 2014**), while the plasma membrane targeting of mPRβ renders the oocyte competent to respond to progesterone (**Nader et al., 2018**).

Exocytosis is also crucial for the formation of the fluid-filled blastocoele cavity during embryogenesis (**Müller, 2001**). Indeed, the apical membrane of the epithelium surrounding the blastula, where the polarized activities of ion channels and transporters generate the blastocoele fluid, is formed from the oocyte cell membrane (**Müller, 2001**). This requires further remodeling. Progesterone rapidly blocks this exocytosis pathway (**Colman et al., 1985**; **Leaf et al., 1990**), leading to a decrease in membrane surface area, which is revealed by the disappearance of microvilli that are enriched in oocytes but almost absent in eggs (**Dumont, 1972**; **Kado et al., 1981**; **Larabell and Chandler, 1989**; **Bluemink et al., 1983**). Importantly, blocking exocytosis induces meiotic maturation, in the absence of hormonal stimulation (**Mulner-Lorillon et al., 1995**; **El Jouni et al., 2007**). Furthermore, progesterone stimulates endocytosis of membrane proteins, such as GPR185, whose activity is suppressed by its internalization, and mPRβ, whose internalization is required to transduce its effects (**Nader et al., 2018**; **Nader et al., 2020**; **Nader et al., 2014**). The stimulation of endocytosis also converts plasma membrane into intracellular vesicles that can provide membrane reserves, necessary to support the rapid cell divisions, and hence the requirement for greater total membrane surface area, during embryogenesis (**Angres et al., 1991**; **Gawantka et al., 1992**). However, it is unknown whether PKA is responsible for the active secretory transport during the prophase arrest and how progesterone blocks exocytosis and stimulates endocytosis (**Figure 1**).

To improve our understanding of vesicular trafficking and membrane organization, we analyzed the proteins involved in these processes whose phosphosites belong to Class I (**Figure 3B**). Among these proteins, PI4KB deserves special attention. This protein is critical for the maintenance of the Golgi and trans-Golgi phosphatidylinositol-4-phosphate (PI4) pools. PI4KB regulates Golgi disintegration/reorganization during mitosis and is involved in Golgi-to-plasma membrane trafficking (**De Matteis et al., 2013**). The phosphoproteome reveals that PI4KB is dephosphorylated within 2hr after progesterone stimulation, on a PKA consensus phosphorylation site conserved in vertebrates. The phosphorylation of this site by PKA is known to control the interaction of PI4KB with Armadillo-like helical domain-containing protein 3 (ARMH3), an interaction important for the Golgi membrane integrity (**McPhail**

*et al., 2020*; *Isobe et al., 2017*; *Blomen et al., 2015*). PI4KB is also regulated by its interaction with ACBD3, an AKAP-like scaffold platform in Golgi, which directly binds the R regulatory subunit of PKA and regulates the traffic between Golgi and endoplasmic reticulum in a PKA-depending manner (*Jia et al., 2023*; *Klima et al., 2016*; *Sasaki et al., 2012*). The phosphoproteome reveals that ACBD3 is phosphorylated during meiotic maturation at the time of NEBD (Class II). PI4KB is, therefore, at the crossroad between PKA and intra-membrane endoplasmic reticulum-Golgi-plasma membrane trafficking. Since these are key events involved in meiosis resumption, PI4KB is a particularly attractive candidate as a PKA substrate for future functional studies of membrane relocalization.

Proteins involved in the actin cytoskeleton are also represented in Class I phosphosites (*Figure 3B*). Their early post-translational modifications could regulate the reorganization of the actin cytoskeleton that is known to accompany the modification of the secretory/endocytosis pathways (*Gard, 1999*). Among them, ITPRID2/KRAP tethers IP$_3$ receptors, which are located in the membrane of the endoplasmic reticulum, to the actin under the plasma membrane, to mediate Ca$^{2+}$ signalling (*Thillaiappan et al., 2021*). During meiotic maturation, the endoplasmic reticulum, which is the major Ca$^{2+}$ store, is enriched in the cortex of the oocyte (*Campanella et al., 1984*; *Charbonneau and Grey, 1984*; *Terasaki et al., 2001*). This remodeling brings the Ca$^{2+}$ source close to its primary targets at fertilization, when Ca$^{2+}$ release from the endoplasmic reticulum activates the oocyte and blocks polyspermy. The proteins highlighted above (*Figure 3B*) could participate in this critical reorganization of the endoplasmic reticulum.

## Protein translation and accumulation orchestrate meiotic divisions

Two waves of translation take place in the oocytes: the first depends on PKA inactivation. It occurs upstream and independently of Cdk1 activity and is required for Cdk1 activation. The second translation wave takes place downstream Cdk1 activation (*Santoni et al., 2024*). Hence, in contrast to mitosis, a period when protein synthesis is repressed (*Ross, 1997*), translation is activated during meiotic maturation, indicating the existence of meiosis-specific controls of the process. However, the regulation of the two waves of protein translation and the identity of the newly translated proteins had not been unraveled. Interestingly, protein translation components are extensively regulated at the level of phosphorylation as demonstrated by the enrichment of the proteins regulating these processes among 3 of the 5 phospho-peptide classes (*Figure 4A*).

### RBPs and translational control

Interestingly, several proteins implicated in the control of translation bear phosphosites belonging to Class I, such as eIF4ENIF1/4E-T, PATL1, and TACC3/Maskin (*Figures 3B and 4B*). eIF4ENIF1/4E-T is required for eIF4e1b localization to P-bodies where mRNAs are stored in a dormant state in zebrafish oocytes (*Lorenzo-Orts et al., 2024*). Both 4E-T and eIF4e1b are extensively phosphorylated on multiple residues during meiotic divisions, suggesting a highly dynamic regulation of P-bodies during this process (*Figure 4B*). Many RBPs known to control translation in oocytes are also regulated at the phosphorylation level, including Pum proteins (*Nakahata et al., 2003*; *Ota et al., 2011*; *Padmanabhan and Richter, 2006*), Zar proteins (*Charlesworth et al., 2012*; *Yamamoto et al., 2013*; *Rong et al., 2019*; *Heim et al., 2022*) and PATL proteins (*Christou-Kent et al., 2018*; *Zhang et al., 2023*; *Marnef et al., 2010*). PATL1 is a RNA-binding protein required for cytoplasmic mRNA P-body assembly in oocytes (*Marnef et al., 2010*). PATL1 and PATL2 have mutually exclusive expression patterns in *Xenopus* oocytes: PATL2 is degraded, as confirmed by our proteomic data (*Figure 2B, D*, *Figure 2—figure supplement 1A–B*), while PATL1 accumulates during meiotic maturation (*Marnef et al., 2010*) or is stable (*Figure 2B*, *Figure 2—figure supplement 1B*). Although the role of PATL2 and the regulation of its stability by phosphorylation have been well documented in the oocyte (*Christou-Kent et al., 2018*; *Zhang et al., 2023*), little is known about the role of PATL1 during meiosis. Both PATL1 and 2 inhibit translation when tethered to mRNA and assemble a complex that includes CPEB1, Xp54/DDX6, Rap55B/LSM14b (*Marnef et al., 2010*; *Figure 4B*). The early dephosphorylation of PATL1 detected in the phosphoproteome could modulate this inhibitory complex during early events of meiotic maturation, while the extensive phosphorylation of the other components of the complex could control its activity later during meiotic divisions (*Figure 4B*). Another component of the repressive complex is TACC3/Maskin. It interacts with CPEB1, although this interaction, and thus its role in regulation of translation, could not be reproduced in other studies (*Minshall et al., 2007*;

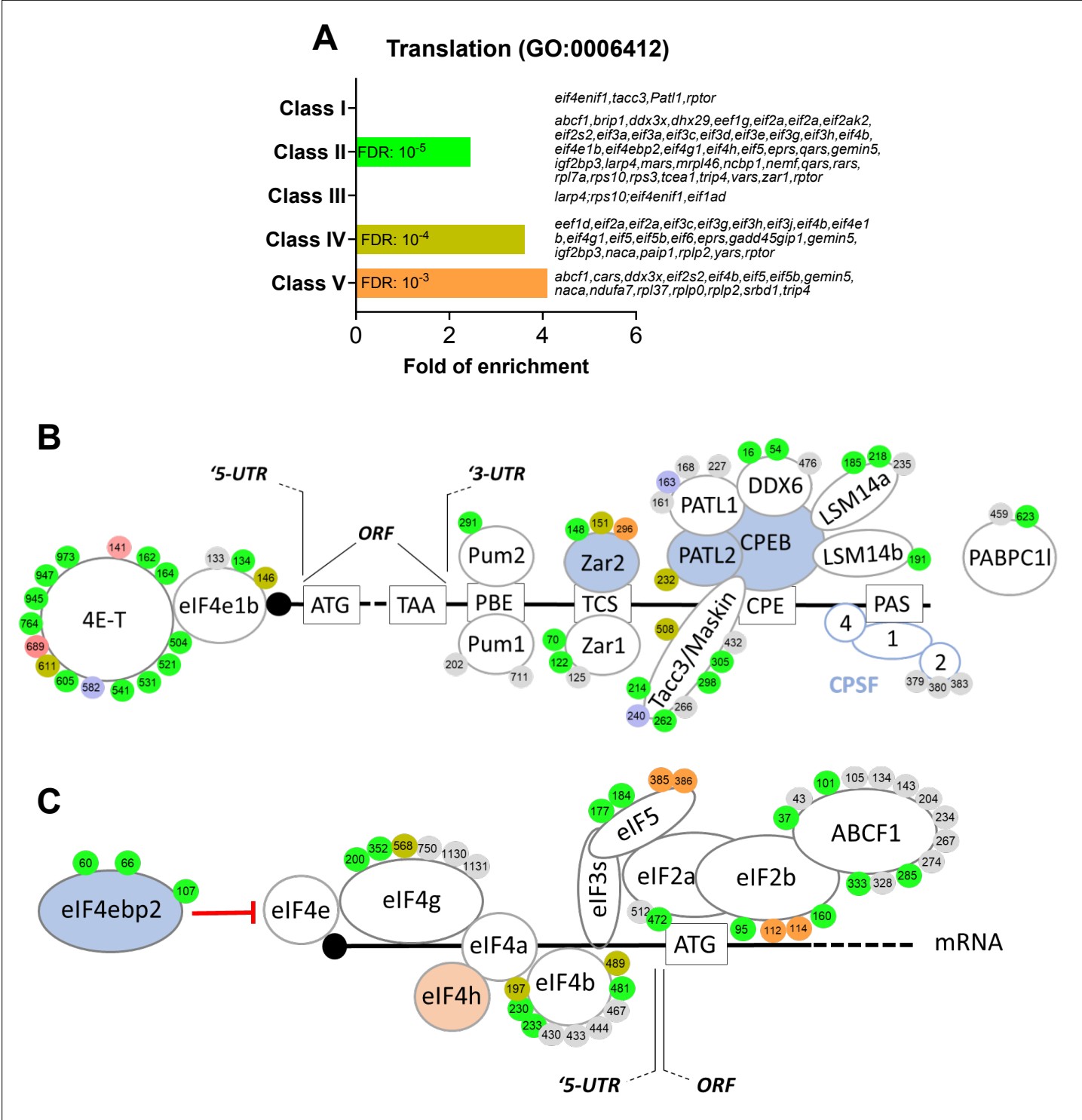

**Figure 4.** RNA-binding proteins and translation initiation factors are dynamically phosphorylated during meiotic divisions. (**A**) Gene ontology enrichment analysis of the biological process 'translation' (GO:0006412) among the phosphopeptides found in each class defined in *Figure 3A*. (**B–C**) Graphical representation of the phosphorylation dynamics of RNA-binding proteins (RBPs) (**B**) and translation initiation factors (**C**) during meiotic divisions. Proteins are color-coded to highlight the changes in protein abundance during meiotic divisions: unchanged = white, decreased = blue, increased = light orange. The phosphorylated sites are color-coded: unchanged, gray; Class I, blue; Class II, green; Class III, red; Class IV, yellow; Class V, orange. The numbering is referred to the iso-allele which has a higher number of phosphorylation sites detected.

The online version of this article includes the following figure supplement(s) for figure 4:

**Figure supplement 1.** Phosphorylation patterns of components of the initiation translation machinery during meiotic divisions.

*Duran-Arqué et al., 2022*). Importantly, besides its function in translation (*Barnard et al., 2005*; *Groisman et al., 2002*), TACC3/Maskin also regulates the centrosome-mediated microtubule nucleation through γ-TuRC (*Peset et al., 2005*). TACC3/Maskin is found widely regulated during meiotic divisions in the phosphoproteome (*Figure 4B*). Eight residues in TACC3/Maskin, of which only two are conserved in human Maskin (214/228 and 298/317 in human/*Xenopus*), exhibit a complex profile of phosphorylation (*Figure 4B*). Two sites identified in the phosphoproteome, S266 and S298, were previously found to be phosphorylated in egg extracts, regulating the function of TACC3/Maskin in the control of translation (*Barnard et al., 2005*). S626 was also reported to be phosphorylated in prophase oocytes and during oocyte meiotic maturation, although there is some controversy about the kinase (either PKA or Aurora-A) and the role of this phosphorylation in either the control of translation or centrosome attachment (*Barnard et al., 2005*; *Pascreau et al., 2005*). Interestingly, in our studies, S240 is phosphorylated in the prophase oocyte and dephosphorylated early in response to progesterone (Class I) (*Figure 4B*). Since the sequence surrounding S240 does not correspond to a PKA consensus motif, it is hard to establish the biological significance; nevertheless, such an early dephosphorylation deserves further attention.

## The control of translation initiation

Translation regulation might not only depend on the expression and post-translational regulation of the RBPs but could also involve the regulation of the core translation initiation machinery, since we find that many of its components are phosphorylated (*Figure 4C*). Indeed, eIF4b, an activator of the eIF4a helicase that facilitates ribosome scanning (*Jackson et al., 2010*), eIF2b/eIF2s2, a component of the molecular machinery recruitment on the met-tRNA (*Jackson et al., 2010*), and eIF4g, involved in the recognition of the mRNA 5'Cap (*Jackson et al., 2010*), are among the proteins bearing sites whose phosphorylation peaks in MII (*Figure 4—figure supplement 1*). These phosphorylation patterns correlate with the second wave of activation of translation that begins under the control of Cdk1 activity in MI and continues progressively through MII (*Meneau et al., 2020*; *Santoni et al., 2024*).

## The control of ubiquitin ligases

The levels of specific proteins are also regulated by proteolysis at each step of meiotic maturation. Targeted regulation allows for the accumulation of key proteins independently of any general increase in their translation, such as for Cyclin B1 (*Santoni et al., 2024*), or the well-studied degradation of proteins, such as Cyclin B2, securin or CPEB1 (*Taieb et al., 2001*; *Wassmann, 2022*; *Setoyama et al., 2007*). Interestingly, proteins controlling polyubiquitination display phospho-peptides enriched in Class II, strengthening the view that a major regulation of protein turnover takes place at NEBD in order to orchestrate meiotic divisions (*Figures 3 and 5A*). APC is the key E3 ubiquitin ligase that promotes metaphase-anaphase transition. Although the phosphorylation status of some substrates determines their recognition by APC and influences the precise timing of their degradation, APC activity is also regulated by the phosphorylation of its core subunits (*Bansal and Tiwari, 2019*; *Figure 5B*).

Another important ubiquitination system that regulates the stability of proteins during mitosis and meiosis is the SCF complex (*Kinterová et al., 2022*). SCF is composed of three main components: Cul1, a scaffold protein; Skp1, which interacts with specific F-box proteins that are involved in the substrate recognition; and Rbx1, which interacts with the E2-ubiquitin ligases (*Kinterová et al., 2022*; *Figure 5C*). SCF ligases play important roles during oocyte meiotic maturation, especially the SCF$^{\beta TrCP}$ complex. Indeed, SCF$^{\beta TrCP}$ mediates the degradation of multiple cell cycle regulators such as Emi1, Emi2/Erp1, CPEB1, and BTG4, whose degradation is required for progression through MI and MII in mouse and *Xenopus* oocytes (*Kinterová et al., 2022*). Similar functions played by such protein degradations occur in mitosis, as Emi1 degradation that releases APC inhibition, then inducing Cyclin B destruction and mitotic exit (*Reimann et al., 2001*).

Many F-Box proteins are also expressed in *Xenopus* oocytes, but they do not display high phosphorylation dynamics. An exception is Lmo7/Fbxo20, which is degraded during meiosis, and is dynamically phosphorylated (*Figure 5C*). Interestingly, Lmo7 overexpression causes defects at the spindle assembly checkpoint, affecting the progression through mitotic divisions (*Tzeng et al., 2018*).

Additionally, many E2 enzymes were identified in our datasets as highly regulated at the translational/accumulation and post-translational levels during meiotic maturation (*Figure 5C*). This layer of

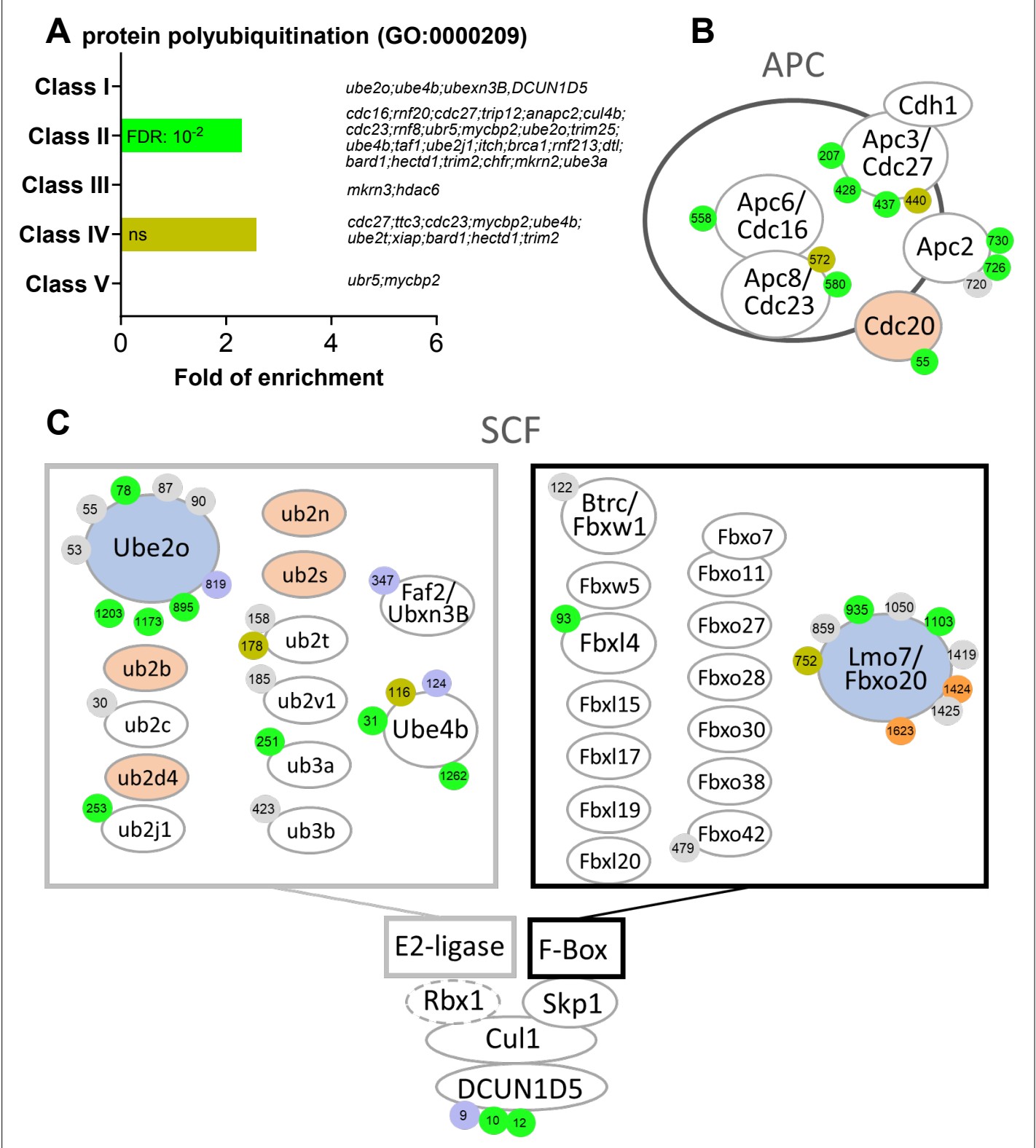

**Figure 5.** E2 and E3 ubiquitin ligases are highly regulated by phosphorylation. (**A**) Gene ontology enrichment analysis of the biological process 'protein polyubiquitination' (GO:0000209) among the phosphopeptides found in each class defined in *Figure 3A*. (**B–C**) Graphical representation of the phosphorylation dynamics of the components of two E3 ubiquitin ligases, the anaphase promoting complex (APC) (**B**) and the Skp, Cullin, F-box containing complex (SCF) (**C**) during meiotic divisions. Proteins are color-coded to highlight changes in protein abundance during meiotic divisions:

*Figure 5 continued on next page*

Figure 5 continued

unchanged = white, decreased = blue, increased = light orange. The phosphorylated sites are color-coded: unchanged, gray; Class I, blue; Class II, green; Class III, red; Class IV, yellow; Class V, orange. The numbering is referred to the iso-allele which has a higher number of phosphorylation sites detected.

regulation of protein ubiquitination remains understudied. Interestingly, an E2/E3 hybrid ubiquitin-protein ligase, UBE2O, UBE4B (also known as UBOX3 or E4), and FAF2 (or Ubxn3B/UBXD8) display phosphorylation sites belonging to Class I (*Figures 3B and 5C*). FAF2 is involved in stress granule clearance (*Gwon et al., 2021*). UBE2O is also known to be involved in the drastic proteome remodeling during erythroid differentiation (*Nguyen et al., 2017*), as well as a quality control factor for orphans of multiprotein complexes (*Yanagitani et al., 2017*). Ribosomal proteins are known substrates of UBE2O which leads to their degradation (*Nguyen et al., 2017*). Whether UBE2O is involved in the downregulation of ribosome biogenesis that occurs during meiotic maturation (*Hyman and Wormington, 1988*), and in the decrease of Rpl26 (*Figure 2—figure supplement 4A*), is an interesting question. The dephosphorylation of UBE2O, UBE4B, and FAF2 could also be involved in the accumulation of Cyclin B1, which was recently shown to occur independently of any increase in translation but to result from protein stabilization (*Santoni et al., 2024*).

Another important regulator of the SCF system, DCUN1D5, is found regulated in our phosphoproteome. DCUN1D5 catalyzes the neddylation of all cullins, which is necessary for the activation of cullin-RING E3 ubiquitin ligases (*Keuss et al., 2016*). DCUN1D5 is dephosphorylated at S9, a PKA consensus site, and phosphorylated starting NEBD at S10 and S12 (*Figure 5C*). These phosphorylations could control DCUN1D5 ability to activate SCF, especially since the inhibition of neddylation causes a meiotic arrest in MI in mouse oocytes (*Yang et al., 2019*). Altogether, our dataset provides attractive candidates to be involved in the regulation of both translation and protein stability that occur during meiotic maturation.

## Activation of Cdk1 occurs through an intricate network of phosphorylation of Cdk1 regulators

The activation of Cdk1 during oocyte maturation relies on an intricate network of feedforward and feedback phosphorylation pathways mediated by kinases and phosphatases (*Figure 1*). Indeed, components of the cell division machinery are highly enriched among Class II, III, IV, and V (*Figure 6A*). Some of the phosphorylation sites implicated in cell division control have been identified previously through detailed studies using site-specific mutagenesis and phospho-specific antibodies. Our phosphoproteomic analysis of oocyte maturation provides new regulatory elements (*Figure 6B*). An important regulator of the Cdk1 auto-amplification loop is the kinase Plk1 that contributes to the phosphorylation of the two direct Cdk1 regulators, Cdc25 and Myt1. Plk1 is activated by Aurora-A, a kinase itself under the indirect control of Cdk1 in *Xenopus* oocyte (*Maton et al., 2003*), through the phosphorylation of T210 (T201 in *Xenopus*) (*Macůrek et al., 2008*), a residue localized in the activation T-loop of the kinase domain. We show here that Plk1 phosphorylation at T210 increases in MI and is then constant throughout meiotic maturation (Class II) (*Figure 6B*, *Figure 6—figure supplement 1A*). This is consistent with the function of Plk1, which phosphorylates multiple proteins required for the mechanics of division, and which must logically be kept active throughout both meiotic divisions (*Hansen et al., 2006*; *Solc et al., 2015*). Interestingly, the phosphoproteome also detects additional phosphorylation sites of Plk1, S326 (S335 in human) and S340 (not conserved in human), which were previously found to be upregulated in response to okadaic acid (*Wind et al., 2002*), an inhibitor of PP2A that strongly induces M-phase. These two residues are located between the two functional domains of Plk1, the kinase domain and the Polo-box domain (*Cheng et al., 2003*). This suggests that the phosphorylation of other critical residues than T210 of Plk1 could be essential for its catalytic activation. Aurora-A kinase is activated during meiotic maturation downstream of Cdk1 (*Maton et al., 2003*; *Castro et al., 2003*; *Maton et al., 2005*). While our phosphoproteome does not detect Aurora-A autophosphorylation at T288 (T295 in *Xenopus*), it registers the extensive phosphorylation of Bora, an activator of Aurora-A involved in Plk1 activation (*Thomas et al., 2016*; *Figure 6B—figure supplement 1B*). Several phosphorylation sites of Bora have been reported to be important for its function as an Aurora-A activator: S41/38, S112/110, S137/135 and S252/S252 (human/*Xenopus*) (*Thomas et al., 2016*; *Vigneron et al., 2018*; *Tavernier et al., 2015*; *Tavernier et al., 2021*). Among

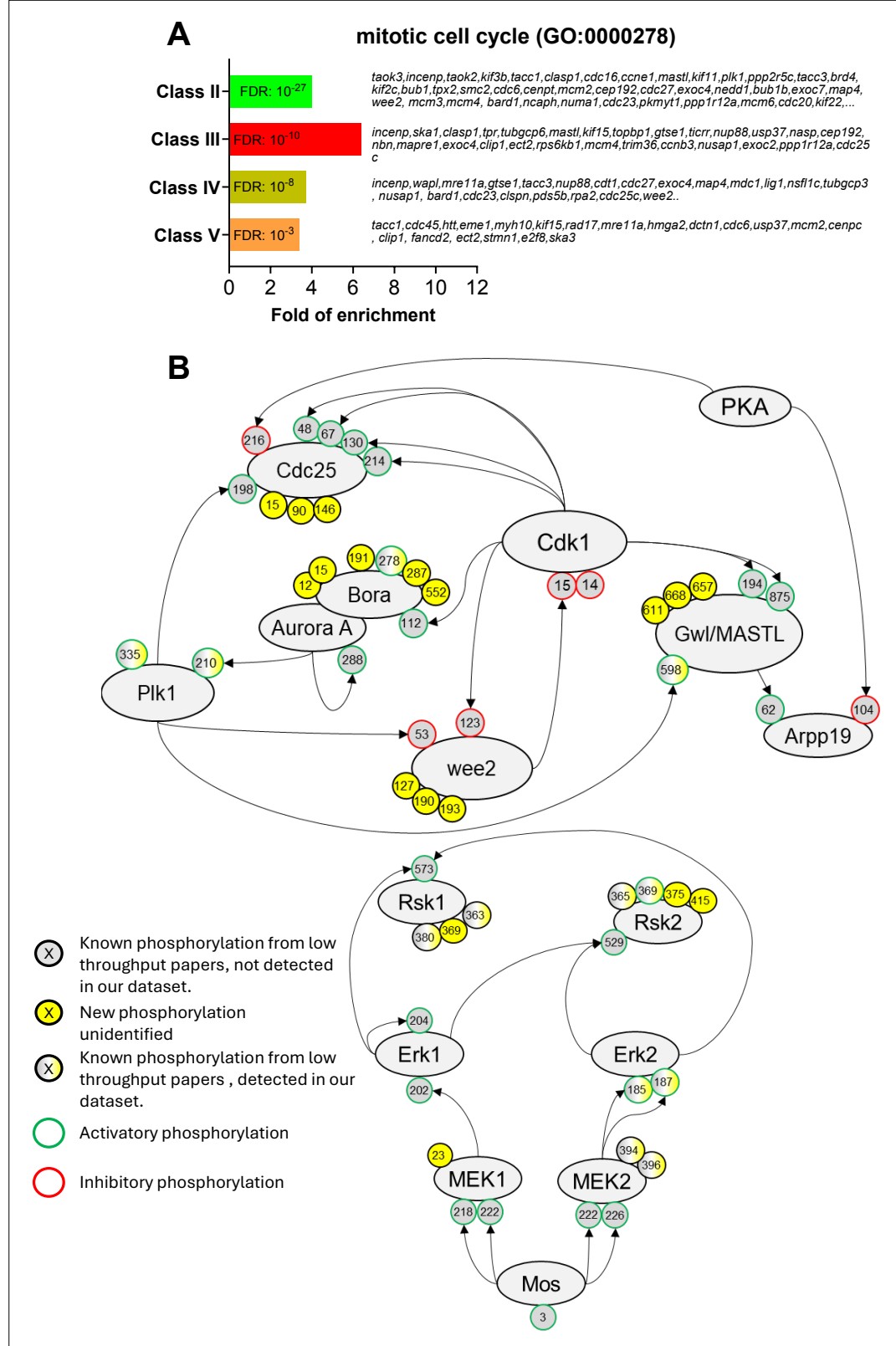

**Figure 6.** The network of phosphorylations regulating Cdk1 and the Mos/MAPK pathway. (**A**) Gene ontology enrichment analysis of the biological process 'mitotic cell cycle' (GO:0000278) among the phosphopeptides found in each class defined in *Figure 3A*. (**B**) Graphical representation of the network of phosphorylations of cell cycle regulators controlling Cdk1 activation (upper panel) and the Mos/MAPK pathway (lower panel). The human

*Figure 6 continued on next page*

*Figure 6 continued*

nomenclature numbering has been used for the phospho-sites. The phosphosites are displayed by dots and color-coded: yellow background, new sites identified in this paper and never described elsewhere; gray background, phosphosites already documented in vertebrates by low throughput papers and not identified in this paper; gray-yellow background, phosphosites already documented in vertebrates by low throughput papers and also identified in this paper. The functional effect of the phosphorylation is indicated by the color of the dot contour line: green, activatory; red, inhibitory; black, unknown.

The online version of this article includes the following figure supplement(s) for figure 6:

**Figure supplement 1.** Phosphorylation patterns of Cdk1 regulators during meiotic divisions.

**Figure supplement 2.** Phosphorylation patterns of the mos/MAPK pathway during meiotic divisions.

these sites, S110 phosphorylation of Bora by Cdk1-Cyclin A is critical for mitotic entry in *Xenopus* egg extracts (*Vigneron et al., 2018*; *Tavernier et al., 2015*; *Tavernier et al., 2021*). Our phosphopro-teome reveals that Bora belongs to Class II, being phosphorylated starting NEBD, but surprisingly on distinct sites that were never reported in the literature: S191/189, T294/287, S552/549, S-/521, T12/9, and T15/12 (human/*Xenopus*), with the exception of S278/285. In contrast to the situation in mitosis, Cyclin A is very weakly expressed in the prophase oocyte and accumulates during meiotic maturation (*Kobayashi et al., 1991*). Our proteome dataset reveals that Cyclin A accumulates after MI. Hence, the phosphorylation of Bora at MI cannot be catalyzed by Cdk1-Cyclin A in the *Xenopus* oocyte. Cdk1-Cyclin B is probably involved in these phosphorylations that take place concomitantly with its activity and at Cdk1 phosphorylation consensus sites for 5 of the 7 phosphosites (*Figure 6B—figure supplement 1B*). Thus, this analysis reveals that Bora activation correlates with its phosphorylation at residues not previously identified in earlier studies of the oocyte. We also detect additional phosphorylation events on other players of the Cdk1 amplification loop, such as Cdc25 and Gwl/MASTL (*Figure 6B—figure supplement 1C–D*), whose extensive phosphorylation during M-phase is usually detected by a large electrophoretic shift (*Dupré et al., 2014*).

## Phosphoregulation of the bimodal activity of Cdk1 during MI-MII transition

In *Xenopus* oocytes, Cdk1 is not fully inactivated at anaphase I and during extrusion of the first polar body. Indeed, the low Cdk1 activity that persists during the MI-MII transition is essential to avoid the reformation of a nucleus, chromosome decondensation and DNA replication (*Huchon et al., 1993*; *Thibier et al., 1997*). This short period is followed by a further increase in Cdk1 activity, due to a sustained synthesis of B-Cyclins; the increased Cdk1 activity allows the formation of the MII spindle (*Gerhart et al., 1984*; *Figure 1*). Although this bimodal subtle regulation of Cdk1 activity is essential for the success of meiosis, i.e., linking two successive divisions without intervening DNA replication, it has never been quantified precisely. Cdk1 activity can be measured by looking at the phosphorylation of one of its direct substrates, Cdc27, a core APC component, at S428 (S426 in human, *Patra and Dunphy, 1998*). Since PP2A, the phosphatase responsible for dephosphorylating Cdc27, is inactive during the MI-MII transition, the phosphorylation levels of Cdc27 can be directly attributed to Cdk1 activity during this period (*Deak et al., 2003*; *Torres et al., 2010*; *Lemonnier et al., 2021*; *Labbé et al., 2021*). Our precise quantification of the phospho-occupancy of this site shows that its phosphorylation is virtually absent in prophase, increases by 40% in MI, decreases during the MI-MII transition to 20%, then increases again in MII by 60% (*Figure 6—figure supplement 1E*). Three other residues of Cdc27, T205/207, S435/437, and S438/440 (human/*Xenopus*), also known to be phosphorylated at M-phase (*Kraft et al., 2003*), follow the same bimodal pattern as S428. The subsequent increase in Cdk1 activity essential for MII entry requires that the kinase escapes inhibitory phosphorylations by Myt1 and Wee2. Both Myt1 and Wee2 are reported to be inhibited by phosphorylations, although the sites responsible for this inhibition are not precisely defined. In *Xenopus*, Wee2 is not expressed in prophase oocytes and accumulates after NEBD (*Nakajo et al., 2000*; *Charlesworth et al., 2012*; *Figure 1*). Our analysis reveals that newly synthesized Wee2 is immediately phosphorylated at four sites (Classes II and IV), one of them being not conserved in human, and none of them corresponding to the proposed inhibitory S53 and S123 sites (*Figure 6*, *Figure 6—figure supplement 1F*). Hence,

these data provide exciting new avenues of research to discover unexpected new regulations of old players in Cdk1 activation.

## The Mos/MAPK pathway activation

Mos plays a well-known role at the origin of a critical pathway controlling meiotic maturation (*Yew et al., 1992*; *Furuno et al., 1994*). It phosphorylates Mek1/2, which in turn phosphorylates Erk1/2 (also known as MAPK) which leads to Rsk1/2 activation (*Figure 6B*). These play multiple essential roles, such as preventing the reformation of a replicating nucleus between MI and MII (*Dupré et al., 2002*), enabling arrest in MII (*Sagata et al., 1989b*), but also participating in Cdk1 activation (*Sagata et al., 1989a*) and the functioning of meiotic spindles (*Araki et al., 1996*; *Choi et al., 1996*; *Bodart et al., 2005*). The phosphoproteome detects the activation of the Mos-MAPK module, with the phosphorylation of Erk2 on T185 and Y187 (T188 and Y190 in *Xenopus*) by MEK1/2 (*Payne et al., 1991*), an event that begins in MI (*Figure 6B—figure supplement 2A*). Interestingly, additional phosphorylation sites are identified in MEK1 and Rsk1/2, the latter being hyperphosphorylated during meiosis as detected by a large electrophoretic shift (*Bhatt and Ferrell, 2000*; *Figure 6B—figure supplement 2B–E*). Rsk1 auto-phosphorylation at S381 (S380 in human) is required for its activity (*Vik and Ryder, 1997*). Our phosphoproteome measures a highly confident phospho-occupancy for this site, demonstrating the switch-like activation of the MAPK pathway at NEBD, followed by the maintenance of a constant level of its activity (*Figure 6B—figure supplement 2F*). The timing of activation of the Mos-MAPK cascade has been controversial, one hypothesis being that it is activated before Cdk1 and participates in the activation of this kinase (*Sagata et al., 1988*; *Gavin et al., 1999*), while another is that it is under the control of the starter Cdk1 activity and is only involved in the autoamplification loop (*Santoni et al., 2024*; *Dupré et al., 2002*; *Fisher et al., 1999*). The phosphoproteomic data clearly support the latter.

## Phosphorylation of key components of the nuclear envelope during breakdown

As a result of the massive activation of kinase activities under the control of Cdk1, hundreds of proteins are phosphorylated and orchestrate the structural events, choreographing the mechanics of meiotic divisions. The first of these is the massive vesiculation of the nuclear envelope surrounding the enormous oocyte nucleus. During mitosis, nuclear pore proteins are extensively phosphorylated by Cdk1-Cyclin B, Plk1, and Nek6/7 to promote the breakdown of the nuclear envelope (*Kutay et al., 2021*). The *Xenopus* oocyte has the unusual feature of having an enormous nucleus (475µm in diameter), and, therefore, a considerable surface area of the nuclear envelope (0.96 mm$^2$), as well as large stocks of annulate lamellae in the cytoplasm. Annulate lamellae are cytoplasmic stack cisternae of nuclear envelope perforated with nuclear pores, 10 times more abundant than in the nucleus in the *Xenopus* oocyte, but devoid of lamins (*Bement and Capco, 1990*; *Miller and Forbes, 2000*; *Cordes et al., 1995*). Their surface area also far exceeds that of the nuclear envelope surrounding the nucleus (*Cordes et al., 1995*). This correlates with the high protein concentrations of the nuclear pore proteins (0.1–0.7µM) that we have measured (*Figure 7A*). Annulate lamellae play a critical role as a reservoir for the formation of the 4000 of nuclei that form during the embryonic cell divisions leading up to the mid-blastula transition and the onset of transcription (*Kessel et al., 1986*; *De Magistris and Antonin, 2018*; *Feldherr, 1974*). During meiotic maturation, NEBD and vesiculation of annulate lamellae occur at the same time, in a progressive spatial manner, starting from the vegetal pole. Importantly, the nucleus and annulate lamellae do not re-form between the two meiotic divisions, reducing the risk that an S-phase can take place between MI and MII (*Bement and Capco, 1990*). A gene ontology analysis of our phosphoproteome highlights that the phosphorylation of nuclear pore components is enriched in Classes II, III, and IV, displaying the highest level of enrichment in Class III, hence characteristic of MI (*Figure 7B*). The nucleoporin Nup53 (also called Nup35) is a component of the soluble core region of the nuclear pore complex that is extensively phosphorylated in mitosis by Cdk1 and Plk1 (*Linder et al., 2017*). Phosphomimetic mutations of all Cdk1 and Plk1 sites in Nup53 slow down the reformation of nuclear pores in interphase and mutation to phospho-null amino acids impairs the nuclear breakdown during meiosis (*Linder et al., 2017*). Our phosphoproteome reveals that Nup53 phosphorylation is more dynamic than previously believed (*Figure 7C*). Indeed, six phospho-peptides are found to peak in MI (Class III), nine phospho-peptides are phosphorylated both in MI and MII

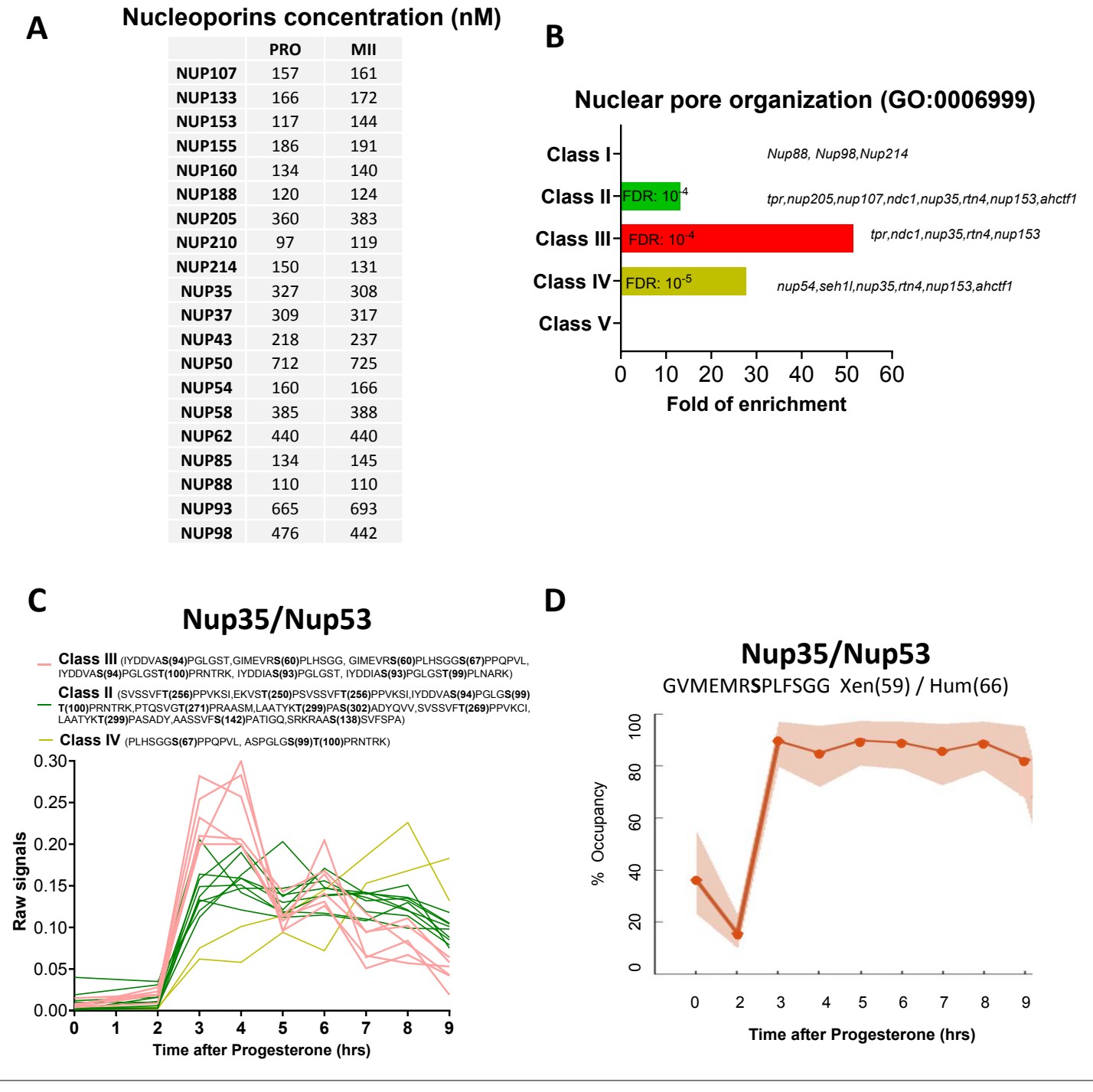

**Figure 7.** The phosphorylation pattern of nuclear pore components during meiotic divisions. (**A**) Absolute concentration (nM) of nucleoporins in prophase (PRO) and metaphase II (MII) oocytes. (**B**) Gene ontology enrichment analysis of the biological process 'nuclear pore organization' (GO:0006999) among the phosphopeptides found in each class defined in *Figure 3A*. (**C**) Phosphorylation patterns of Nup35/Nup53. The phosphorylation curves are color-coded: Class II, green; Class III, red; Class IV, yellow. (**D**) High precision phospho-occupancy calculation of S59 phosphorylation pattern of Nup35/Nup53.

(Class II), and two phospho-peptides continuously increase in phosphorylation during meiotic maturation (Class IV) (*Figure 7C*). We were able to calculate the phospho-occupancy of the S59 (S66 in human), showing that it increases from 30% in prophase to 90% in MI oocytes and stays constant until MII (Class II) (*Figure 7D*). Several other transmembrane nucleoporins, notably NDC1 and POM121C,

undergo a dramatic increase in phosphorylation at the transition to MI. Altogether, these observations indicate that dissociation of the nuclear envelope and the annulate lamellae occurring in MI might require either a specific set of phosphorylations and/or a general level of phosphorylation that is higher than the level of phosphorylation required to maintain the nucleus and the annulate lamellae dissociated during the MI-MII transition through MII.

## The regulation of centrosomal proteins during oocyte maturation

An intriguing feature of oocytes in most, if not all, metazoan species is the elimination of their centrioles. This allows the sperm to contribute its centriole and thereby initiates the division cycle and the cell cycle. There is longstanding literature that proves that the oocyte or egg has retained its ability to generate centrioles. However, in nature, the female centrioles are lost and the sperm centriole is the main contributor. In *Xenopus*, oocytes entering prophase I contain a typical centrosome, including a pair of centrioles and pericentriolar material (PCM) containing γ-tubulin and pericentrin. Centrioles disappear at the end of the pachytene stage of prophase I and PCM disperses in multiple foci in the cytoplasm that are no longer functional as microtubule nucleation sites (*Gard et al., 1995*). At fertilization, the male gamete delivers two centrioles but does not provide the PCM (*Delattre and Gönczy, 2004*). Therefore, in most species (with the exception of rodents), the complete centrosome of the one-cell stage embryo is a bi-parental inheritance, reconstituted from paternal centrioles and maternal PCM (*Delattre and Gönczy, 2004*). In *Xenopus*, oocytes store enough quiescent centrosomal building blocks for duplicating and assembling 1000–2000 centrosomes (*Gard et al., 1990*). This implies that oocytes should comprise an enormous reserve of centrosomal proteins, i.e., centriolar and PCM proteins, that sustain the reconstitution of a functional centrosome at fertilization and its subsequent rounds of duplication in the early embryo.

### Centriolar proteins

The centriole is formed by a polarized cylinder of microtubule triplets, decorated along the proximal to distal axis with several appendages that define the three main regions of the organelle: the proximal part, the central core, and the distal region (*LeGuennec et al., 2021*). In agreement with a published RNA-seq and proteome of *Xenopus* eggs (*Session et al., 2016*; *Wühr et al., 2014*), our proteome reveals that most of the centriolar proteins are expressed in the oocyte (*Figure 8*). The components of the proximal part, such as SAS-6 and CPAP/CENPJ, and of the central core, as POC1B, POC5, and Centrin 1, are either constant or accumulate during meiotic maturation (*Figure 8*). In contrast, proteins of distal and subdistal appendages are present at lower concentrations, either constant or decreasing during meiotic maturation (*Figure 8*). Indeed, the appendages are critical for the docking of mature centrioles at the plasma membrane and ciliogenesis (*Tanos et al., 2013*), a process that is not functional in the oocyte and during the early embryonic divisions (*Drysdale and Elinson, 1992*).

Besides the basic building blocks forming the centriole, the oocyte is expected to provide the machinery for its duplication, the first cycle occurring immediately after fertilization. The key regulators of centriole duplication, including Plk4, SAS-6, and STIL, are generally expressed at very low levels (*Bauer et al., 2016*). While Plk4 is expressed at the mRNA level in oocytes but escaped our proteomic detection, SAS-6 and STIL are expressed at a constant level during meiotic maturation (*Figure 8*). The estimated concentrations of both proteins are in agreement with the previous predictions, that cartwheels comprise one molecule of STIL for every SAS-6 dimer (*Bauer et al., 2016*). Other important regulators of centriole duplication, such as CEP192, CEP152, CEP63, RTTN, CEP110/CCP110, CEP97, and CEP76 were also detected, and are either constant or accumulate during meiotic divisions (*Figure 8*). Interestingly, our phosphoproteome reveals that several of these proteins are phosphorylated during meiotic maturation. STIL is phosphorylated at S554, which is located in a Cdk1 phosphorylation consensus (*Figure 8—figure supplement 1A*), an interesting observation since phosphorylation of STIL by Cdk1-Cyclin reduces the efficiency of daughter centriole assembly. This phosphorylation provides an explanation for why centrioles cannot duplicate during M-phase (*Zitouni et al., 2016*; *Steinacker et al., 2022*). A massive CEP97 phosphorylation occurs at MI at 8 distinct sites, only 3 of them lying within a Cdk1 consensus motif (*Figure 8—figure supplement 1B*). Remarkably, one of the phosphosites of CEP97, S633, has been recently shown to be phosphorylated by the kinase Dyrk1a in multiciliated cells of *Xenopus* embryos (*Lee et al., 2022*); it is required for centriole duplication. Therefore, although centrioles are not assembled, all centriolar proteins are expressed

| | Gene name | Proteome | | | | | | | | | | (Wuhr,2014) | (Session,2016) |
|---|---|---|---|---|---|---|---|---|---|---|---|---|---|
| | | 0 | 2 | 3 | 4 | 5 | 6 | 7 | 8 | 9 | Log2 FC (MII/PRO) | Estimated concentration in MII-arrested oocytes (nM) | Count per Million (CPM) in PRO I |
| **Proximal part** | SAS-6 | 0.11 | 0.09 | 0.10 | 0.10 | 0.09 | 0.09 | 0.09 | 0.09 | 0.08 | -0.3 | 36 | 20 |
| | CENPJ | 0.07 | 0.09 | 0.09 | 0.10 | 0.10 | 0.11 | 0.09 | 0.10 | 0.09 | 0.4 | 9 | 56 |
| | Cep135 | not detected | | | | | | | | | | 5 | 1 |
| **Central core** | POC1B | 0.05 | 0.05 | 0.07 | 0.08 | 0.09 | 0.14 | 0.11 | 0.10 | 0.11 | 1.2 | 7 | 96 |
| | POC5 | 0.05 | 0.06 | 0.07 | 0.08 | 0.09 | 0.10 | 0.10 | 0.11 | 0.12 | 1.1 | 39 | 100 |
| | CETN1 | 0.08 | 0.08 | 0.11 | 0.09 | 0.09 | 0.09 | 0.10 | 0.09 | 0.10 | 0.2 | not detected | 3 |
| | SFI1 | not detected | | | | | | | | | | 3 | 7 |
| | POC1A | not detected | | | | | | | | | | 20 | 54 |
| | CETN2 | not detected | | | | | | | | | | 82 | not detected |
| | WDR90 | not detected | | | | | | | | | | 5 | 25 |
| | FAM161A | not detected | | | | | | | | | | not detected | 34 |
| **Distal appendages** | SCLT1 | 0.10 | 0.09 | 0.10 | 0.11 | 0.10 | 0.10 | 0.08 | 0.08 | 0.08 | -0.4 | 13 | 2 |
| | FBF1 | 0.11 | 0.12 | 0.11 | 0.12 | 0.11 | 0.11 | 0.08 | 0.07 | 0.06 | -0.7 | not detected | 17 |
| | CEP83 | not detected | | | | | | | | | | not detected | 7 |
| | CEP89 | not detected | | | | | | | | | | not detected | 1 |
| **Sub-distal appendages** | FBF1 | 0.11 | 0.12 | 0.11 | 0.12 | 0.11 | 0.11 | 0.08 | 0.07 | 0.06 | -0.7 | not detected | 17 |
| | CEP164 | 0.10 | 0.08 | 0.11 | 0.09 | 0.12 | 0.09 | 0.08 | 0.06 | 0.08 | -0.4 | 1 | not detected |
| | NIN | 0.11 | 0.09 | 0.11 | 0.07 | 0.08 | 0.11 | 0.10 | 0.08 | 0.10 | -0.2 | 4 | 27 |
| | TUBE1 | 0.10 | 0.09 | 0.10 | 0.10 | 0.10 | 0.11 | 0.09 | 0.09 | 0.09 | -0.1 | 7 | 8 |
| | TUBD1 | 0.08 | 0.08 | 0.08 | 0.09 | 0.09 | 0.11 | 0.09 | 0.09 | 0.10 | 0.1 | 13 | 28 |
| | ODF2 | not detected | | | | | | | | | | 7 | 8 |
| | CEP128 | not detected | | | | | | | | | | 3 | 17 |
| | CNTRL | not detected | | | | | | | | | | 6 | 26 |
| | CEP170 | not detected | | | | | | | | | | 4 | 3 |
| | TCHP | not detected | | | | | | | | | | 2 | 16 |
| | CCDC68 | not detected | | | | | | | | | | not detected | 3 |
| | tubz1 | not detected | | | | | | | | | | not detected | 10 |
| | CCDC120 | not detected | | | | | | | | | | not detected | not detected |
| **Centriole duplication** | CCP110 | 0.06 | 0.07 | 0.06 | 0.08 | 0.08 | 0.09 | 0.10 | 0.12 | 0.12 | 0.9 | 7 | 20 |
| | CEP152 | 0.07 | 0.07 | 0.10 | 0.10 | 0.10 | 0.12 | 0.10 | 0.11 | 0.09 | 0.6 | 7 | 161 |
| | CEP192 | 0.08 | 0.08 | 0.08 | 0.08 | 0.09 | 0.09 | 0.10 | 0.11 | 0.10 | 0.5 | 21 | 24 |
| | STIL | 0.08 | 0.08 | 0.07 | 0.08 | 0.10 | 0.11 | 0.10 | 0.11 | 0.12 | 0.5 | 14 | 100 |
| | CEP63 | 0.07 | 0.06 | 0.11 | 0.11 | 0.13 | 0.12 | 0.10 | 0.09 | 0.09 | 0.5 | 4 | 33 |
| | CEP76 | 0.07 | 0.07 | 0.07 | 0.08 | 0.09 | 0.14 | 0.10 | 0.07 | 0.12 | 0.5 | 12 | 57 |
| | RTTN | 0.09 | 0.09 | 0.09 | 0.10 | 0.10 | 0.11 | 0.09 | 0.09 | 0.08 | 0.0 | 18 | 14 |
| | CEP97 | 0.10 | 0.10 | 0.10 | 0.10 | 0.09 | 0.10 | 0.10 | 0.09 | 0.09 | -0.1 | 15 | 1 |
| | SAS-6 | 0.11 | 0.09 | 0.10 | 0.10 | 0.09 | 0.09 | 0.09 | 0.09 | 0.08 | -0.3 | 36 | 42 |
| | Plk4 | not detected | | | | | | | | | | not detected | not detected |
| **PCM** | KIF22 | 0.02 | 0.02 | 0.04 | 0.05 | 0.07 | 0.09 | 0.12 | 0.14 | 0.19 | 3.1 | 33 | 464 |
| | SSX2IP | 0.03 | 0.03 | 0.05 | 0.07 | 0.09 | 0.11 | 0.11 | 0.13 | 0.16 | 2.1 | 20 | 10 |
| | TUBG1 | 0.04 | 0.04 | 0.04 | 0.04 | 0.04 | 0.04 | 0.12 | 0.12 | 0.11 | 1.5 | 166 | 101 |
| | tpx2 | 0.06 | 0.06 | 0.07 | 0.08 | 0.10 | 0.11 | 0.11 | 0.12 | 0.11 | 1.0 | 43 | 516 |
| | kif11 | 0.06 | 0.06 | 0.07 | 0.08 | 0.09 | 0.10 | 0.10 | 0.11 | 0.12 | 0.9 | 44 | 185 |
| | dync1li1 | 0.06 | 0.0735 | 0.0605 | 0.0675 | 0.062 | 0.0605 | 0.104 | 0.1185 | 0.087 | 0.8 | 169 | 38 |
| | dync2h1 | 0.067 | 0.057 | 0.073 | 0.074 | 0.092 | 0.098 | 0.104 | 0.105 | 0.128 | 0.7 | 35 | 3 |
| | prc1 | 0.06 | 0.07 | 0.08 | 0.09 | 0.10 | 0.12 | 0.09 | 0.10 | 0.10 | 0.7 | 32 | 13 |
| | CEP152 | 0.07 | 0.07 | 0.10 | 0.10 | 0.10 | 0.12 | 0.10 | 0.11 | 0.09 | 0.6 | 7 | 33 |
| | kif4a | 0.08 | 0.07 | 0.08 | 0.09 | 0.09 | 0.09 | 0.10 | 0.10 | 0.10 | 0.4 | 100 | 120 |
| | TUBGCP4 | 0.08 | 0.08 | 0.09 | 0.09 | 0.10 | 0.08 | 0.10 | 0.10 | 0.10 | 0.2 | 14 | 20 |
| | nedd1 | 0.08 | 0.08 | 0.09 | 0.08 | 0.09 | 0.08 | 0.10 | 0.10 | 0.09 | 0.2 | 30 | 12 |
| | dync2i1 | 0.0875 | 0.076 | 0.0915 | 0.088 | 0.0965 | 0.0975 | 0.0925 | 0.0935 | 0.101 | 0.1 | not detected | 11 |
| | CKAP5 | 0.09 | 0.09 | 0.09 | 0.09 | 0.09 | 0.10 | 0.09 | 0.10 | 0.09 | 0.1 | 67 | 102 |
| | dynll2 | 0.086 | 0.111 | 0.077 | 0.102 | 0.09 | 0.08 | 0.086 | 0.085 | 0.104 | 0.1 | 411 | 60 |
| | dync2i2 | 0.089 | 0.081 | 0.1 | 0.099 | 0.106 | 0.098 | 0.09 | 0.087 | 0.095 | 0.0 | not detected | 35 |
| | TUBGCP3 | 0.09 | 0.08 | 0.09 | 0.09 | 0.10 | 0.10 | 0.09 | 0.10 | 0.09 | 0.0 | 33 | 10 |
| | KATNB1 | 0.09 | 0.09 | 0.10 | 0.10 | 0.10 | 0.10 | 0.09 | 0.09 | 0.09 | 0.0 | 20 | 108 |
| | dynlt1 | 0.086 | 0.1 | 0.114 | 0.09 | 0.09 | 0.07 | 0.083 | 0.085 | 0.093 | 0.0 | 138 | 49 |
| | dync1i2 | 0.091 | 0.104 | 0.104 | 0.073 | 0.081 | 0.106 | 0.091 | 0.089 | 0.092 | 0.0 | 232 | 48 |
| | PCNT | 0.09 | 0.09 | 0.09 | 0.10 | 0.10 | 0.10 | 0.09 | 0.09 | 0.09 | 0.0 | 16 | 23 |
| | NPM2 | 0.08 | 0.10 | 0.08 | 0.12 | 0.08 | 0.08 | 0.07 | 0.07 | 0.09 | 0.0 | 1181 | 116 |
| | dync2li1 | 0.086 | 0.074 | 0.084 | 0.086 | 0.089 | 0.091 | 0.082 | 0.081 | 0.091 | 0.0 | 43 | 4 |
| | CLASP1 | 0.09 | 0.08 | 0.09 | 0.10 | 0.10 | 0.10 | 0.09 | 0.09 | 0.08 | -0.1 | 15 | 48 |
| | TUBGCP2 | 0.09 | 0.09 | 0.10 | 0.10 | 0.10 | 0.09 | 0.09 | 0.09 | 0.09 | -0.1 | 59 | 22 |
| | dync1li2 | 0.097 | 0.089 | 0.099 | 0.104 | 0.094 | 0.099 | 0.091 | 0.094 | 0.086 | -0.1 | 26 | 27 |
| | dynll1 | 0.099 | 0.101 | 0.094 | 0.096 | 0.092 | 0.105 | 0.089 | 0.093 | 0.089 | -0.1 | 51 | 130 |
| | NEK9 | 0.09 | 0.10 | 0.10 | 0.10 | 0.10 | 0.10 | 0.09 | 0.08 | 0.08 | -0.2 | 45 | 27 |
| | TACC3 | 0.10 | 0.10 | 0.10 | 0.10 | 0.10 | 0.10 | 0.09 | 0.09 | 0.08 | -0.2 | 115 | 117 |
| | TUBGCP6 | 0.10 | 0.09 | 0.11 | 0.10 | 0.10 | 0.10 | 0.08 | 0.08 | 0.08 | -0.3 | 8 | 14 |
| | NPM3 | 0.11 | 0.09 | 0.12 | 0.12 | 0.11 | 0.10 | 0.09 | 0.09 | 0.08 | -0.3 | 150 | 70 |
| | KATNA1 | 0.10 | 0.10 | 0.10 | 0.09 | 0.10 | 0.09 | 0.09 | 0.08 | 0.08 | -0.3 | 63 | 30 |
| | NPM1 | 0.12 | 0.09 | 0.12 | 0.10 | 0.09 | 0.09 | 0.08 | 0.08 | 0.08 | -0.6 | 308 | 102 |
| | TUBGCP5 | not detected | | | | | | | | | | 24 | not detected |
| | KATNAL1 | not detected | | | | | | | | | | 5 | not detected |
| | KATNAL2 | not detected | | | | | | | | | | not detected | 22 |

**Figure 8.** Centrosomal components during meiotic divisions. Expression patterns of the components of the centrioles and the pericentriolar material. The changes in protein abundance are color-coded: unchanged = white, decreased = blue, increased = light orange. The estimation of the protein and mRNA concentrations was derived from published datasets and color-coded on a white-green scale for protein concentrations and on a white-yellow scale for mRNA concentrations.

*Figure 8 continued on next page*

*Figure 8 continued*

The online version of this article includes the following figure supplement(s) for figure 8:

**Figure supplement 1.** Phosphorylation patterns of centrosomal proteins during meiotic divisions.

in the full-grown oocyte and these components serve as a stock for the biogenesis of the centrioles of the embryo. Our results suggest that specific phosphorylations of proteins regulating centriole assembly could prevent the de novo establishment of centrioles in the oocyte.

## PCM proteins

The core element of the PCM is the γ-tubulin ring complex (γ-TuRC) which is the basic element for nucleation of microtubules from microtubule-organizing centers (MTOCs) (*Oakley and Oakley, 1989*). It includes γ-tubulin that is combined with other proteins known as GCP2/tubgcp2 to GCP6/tubgcp6. Some PCM proteins or regulators of PCM assembly have been described in functional studies to be present in the oocyte and then recruited to the zygotic centrosome as maternal components. These include γ-tubulin (*Stearns and Kirschner, 1994*; *Gard, 1994*; *Félix et al., 1994*), pericentrin (*Dictenberg et al., 1998*), Nek2B (*Fry et al., 2000*; *Uto and Sagata, 2000*), SSX2IP (*Bärenz et al., 2013*), TPX2, PRC1, Kif4A, Eg5/Kif11, CLASP1/Xorbit and Kif22/Xkid, CEP152 (*Hatch et al., 2010*), NEDD1 (*Liu and Wiese, 2008*), Maskin/TACC3 (*Stebbins-Boaz et al., 1999*; *Peset et al., 2005*), XMAP215/CKAP5 (*Popov et al., 2002*; *Kinoshita et al., 2002*), and Nercc1/NEK9 (*Roig et al., 2005*).

Most of the PCM proteins were detected in our proteomic dataset and either accumulate or stay constant during meiotic divisions (*Figure 8*). Our dataset highlights that γ-tubulin accumulates during meiotic divisions at the time the oocytes enter MII (*Figure 8—figure supplement 1C*). GCP2 and 4 are expressed at a lower level than γ-tubulin (*Figure 8*), in agreement with the expected stoichiometry of GCPs-γ-tubulin in the γ-TuRC (*Sulimenko et al., 2022*). The phosphorylation of γ-TuRC components is known to affect the stability and the activation of the complex (*Sulimenko et al., 2022*). We detected the phosphorylation of GCP3 at S814 (Class IV) and GCP6 at S1173 (Class III) (*Figure 8—figure supplement 1D*). This second phosphosite is well conserved in the human protein and targeted by Plk4, which regulates centriole duplication (*Bahtz et al., 2012*). These results indicate that the oocyte, despite the absence of centrioles, prepares a functional centrosomal machinery by accumulating and/or phosphorylating some of the PCM proteins. Apart from the γ-TuRC complex, we found that 10 other PCM proteins are phosphorylated during meiotic maturation (CEP192, SSX2IP, CEP41, TACC3/Maskin, Nucleophosmin, Katanin, Kif11, Kif15/Eg5, Numa, and Dynein). CEP192 is a coiled-coil scaffolding protein that recruits γ-TuRC, Aurora-A, and Plk1 to the centrosome for its maturation. Our phosphoproteome reveals that CEP192, a Class II protein, is phosphorylated at seven sites between prophase and MI, five of them corresponding to Cdk1 phosphorylation sites (*Figure 8—figure supplement 1E*). Plk1 is recruited to its many different locations in the cell through its Polo-Box-Domain (PBD), which binds to phosphorylated S-S/T(p) motifs (*Elia et al., 2003a*; *Elia et al., 2003b*; *Lee et al., 1998*; *Reynolds and Ohkura, 2003*). Two of the CEP192 sites identified in our phosphoproteome, S991, and S1227, correspond to such motifs. Therefore, by attracting Plk1, CEP192 phosphorylation could prepare the reconstitution of a functional centrosome at fertilization.

## Assembly of microtubule spindles and microtubule dynamics

*Xenopus* prophase oocytes are inefficient in their ability to support microtubule polymerization (*Jessus et al., 1984*; *Gard and Kirschner, 1987*; *Heidemann and Kirschner, 1975*), although microtubules are formed at the cortical and nuclear envelope level, where γ-tubulin foci are also localized (*Gard, 1994*; *Jessus et al., 1988*). The ability to assemble microtubule asters is acquired at the time of NEBD, when a giant disk-shaped MTOC generates a large microtubular monoaster at the basal region of the disintegrating nucleus (*Huchon et al., 1981*; *Gard, 1994*). Therefore, MTOC and microtubule-associated proteins (MAPs) must be finely regulated by post-translational modifications at the time of NEBD. The transient microtubule monoaster transports the meiotic chromosomes to the animal cortex, where the first meiotic spindle is assembled (*Huchon et al., 1981*; *Gard, 1992*). Due to the absence of centrioles, meiotic spindles are barrel-shaped and are formed by centrosome-independent microtubule nucleation originating from the chromosomes and involving chromatin and the Ran GTPase (*Gruss and Vernos, 2004*), much like spindles of vascular plant cells. Moreover, the

two successive spindles are differentially regulated. The MI spindle is a transient and dynamic structure, while the MII spindle is stable for hours, anchored to the plasma membrane until fertilization (*Gard, 1992*). Therefore, it is possible that MAPs are specifically and differentially regulated at MI and MII (Classes III and V). Finally, a major change in microtubule assembly occurs during the transition from meiosis to embryonic mitosis. The assembly of the first embryonic spindle integrates the meiotic chromosome-dependent pathway with the newly formed centrosomes (*Cavazza et al., 2016*). Thereafter, subsequent embryonic spindles depend solely on the microtubule nucleating activity of the

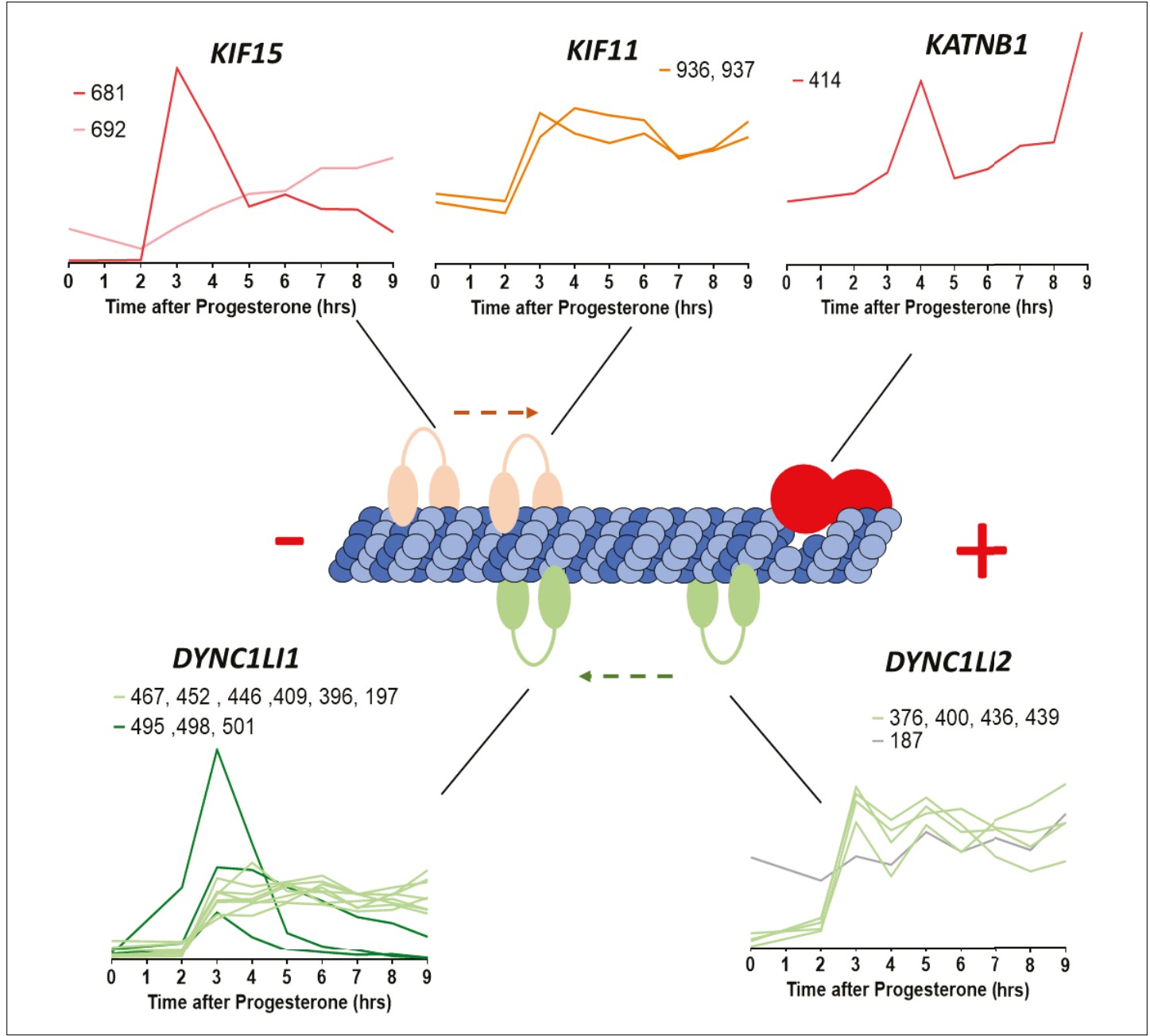

**Figure 9.** Microtubule-interacting proteins phosphorylation during meiotic divisions. Graphic representation of the phosphorylation dynamics of kinesins (Kif15, Kif11), dyneins (DYNC1LI1, DYNC1LI2) and katanin (KATNB1) during meiotic divisions. The sites of the phosphorylation detected are marked above each panel.

The online version of this article includes the following figure supplement(s) for figure 9:

**Figure supplement 1.** Phosphorylation patterns of cohesins during meiotic divisions.

centrosomes (*Kirschner, 1986*). All these critical transitions in the way of assembling a microtubular spindle depend on the proteins stored in the oocyte.

Many regulators of the microtubular spindle are highly regulated during meiotic divisions, both at the level of their accumulation (*Figure 2B–C*) and by post-translation modifications (*Figure 9*). At least five microtubule regulatory proteins undergo specific patterns of phosphorylation that deserve attention (*Figure 9*). Katanin (KATNB1), which is responsible for the majority of M-phase severing activity in *Xenopus* eggs and is activated by Cdk1 (*McNally and Thomas, 1998*), displays a bi-modal pattern of phosphorylation, increasing at the time of formation of the monoaster-MTOC and MI spindle, decreasing during MI-MII transition, and increasing again at MII (*Figure 9*). Kif11/Eg5 phosphorylation occurs at T936 and T937 starting NEBD (*Figure 9*). Both phosphosites were reported in a broad range of models, and the second site was shown to be phosphorylated by Cdk1 (*Blangy et al., 1995*; *Cahu et al., 2008*; *Giet et al., 1999*; *Figure 9*). Kif11/Eg5 controls the relative stability of bipolar versus monopolar organization of spindles in *Xenopus* egg extracts (*Mitchison et al., 2005*) but also centrosome disjunction and/or separation. Hence, Kif11 is a good candidate to be an important player regulating the original changes in microtubule dynamics and organization of the two successive spindles. Kif15, also known as XKlp2, a plus end-directed kinesin protein required for centrosome separation and maintenance of spindle bipolarity in *Xenopus* egg extracts (*Boleti et al., 1996*), is phosphorylated at S681 and S692 during meiotic maturation. S681 is rather phosphorylated at MI, while S692 is specific to MII (*Figure 9*), suggesting a potential distinct regulation of Kif15 on microtubule dynamics at MI and MII. In contrast to Kif11/Eg5 and Kif15, some kinesins from the kinesin-14 family, such as Kifc1, whose protein level increases during meiotic divisions (*Figure 2C*), as well as dyneins, are motor proteins that move to microtubules in the opposite direction to kinesins. Among the dynein family, the two paralog genes dynein light intermediate chain (DynC1LI1 and DynC1LI2) play important roles in mitosis, including positioning the spindle and focusing the MTs into poles. Our phosphoproteome reveals multiple phosphorylations of these proteins, starting in MI and remaining constant or progressively increasing until MII, as for S197 of DynC1LI1 (S207 in human) (*Figure 9*). This agrees with previous data showing that S197 is phosphorylated by Cdk1 (*Dell et al., 2000*; *Addinall et al., 2001*) and that DynC1LI becomes hyperphosphorylated at the time of NEBD and remains hyperphosphorylated throughout the rest of meiosis (*Huang et al., 1999*). Additionally, our phosphoproteome reveals that three sites (495, 498, and 501) are phosphorylated around NEBD and get dephosphorylated after MI (Class III), hence being correlated with the first meiotic division and not the second one, while six other sites (197, 396, 409, 446, 452, and 467) are phosphorylated both in MI and MII (Class II) (*Figure 9*). Dynein is required for the organization of the original microtubule/MTOC array that is organized at NEBD, together with XMAP215, XKCM1, and Numa (*Becker et al., 2003*). The formation of this original oocyte monoaster/MTOC could involve the dynein phosphorylations that transiently take place precisely at this period.

Strikingly, MI and MII spindles segregate chromosomes and chromatids in a very different way. In MI, the homologous chromosomes are segregated, whereas in MII the sister chromatids are displaced in the daughter cells, due to a specific organization of kinetochores and regulation of cohesins. In MI, arm cohesins are phosphorylated and degraded, whereas centromeric cohesins are protected from cleavage by Sugoshin (Sgo1)-PP2A. This allows the separation of chromosomes, but chromatids are still attached together. In MII, centromeric cohesins are fully degraded, allowing the segregation of sister chromatids, as during mitosis (*Wassmann, 2022*). How a fraction of cohesins is protected from cleavage in MI but not in MII is still not entirely clear since the phosphorylation of cohesins and their partners, which are not very abundant proteins, is difficult to detect with conventional shotgun phosphoproteomics. Notably, our phosphoproteomic studies highlight new sites differentially phosphorylated between MI and MII within these proteins. Sgo1 and two components of the cohesin complex, Pds5b and Wapl, are increasingly phosphorylated during meiotic divisions (Class IV) (*Figure 9—figure supplement 1*). Noteworthy, S1069 of Wapl is a putative Plk1 site (*Grosstessner-Hain et al., 2011*; *Kettenbach et al., 2011*). By highlighting new sites differentially phosphorylated between MI and MII, our phosphoproteomic studies provide clues to elucidate the still unknown mechanisms of the meiotic mechanism of cohesion protection-deprotection.

## Evolutionary conservation of the phospho-proteome between *Xenopus* and mouse

We compared our phosphoproteome to a dataset obtained during meiotic divisions in mouse oocytes (*Sun et al., 2024*). We identified 408 phosphorylation sites present in both datasets (see Methods) (*Supplementary file 5*). These sites correspond to 320 *Xenopus* proteins and 277 mouse proteins. Note that when multiple mouse proteins share the same phospho-peptide sequence, they are grouped together in the mouse dataset. However, alignment with *Xenopus* data allows us to resolve which specific mouse protein is phosphorylated, thereby enriching the mouse dataset by leveraging information from *Xenopus* measurements. We then assessed whether the changes in phosphorylation between prophase and metaphase II were consistent between *Xenopus* and mouse meiotic maturation (*Figure 10A*). Interestingly, the phosphorylation dynamics were significantly correlated between the two datasets (Pearson coefficient: 0.39, p<0.0001) (*Figure 10A*). Furthermore, the changes in phosphorylation observed in *Xenopus* oocytes (range of the $\text{Log}_2$ Fold Change (MII/Pro): –4.27 and 7.03) were substantially greater than those measured in the mouse dataset (range of the $\text{Log}_2$Fold Change (MII/Pro): –2.55 and 3.72), likely reflecting the higher dynamic range achieved by our workflow. Among the key phosphorylation events detected in both datasets are those involved in the activation of Cdk1 and the Mos/MAPK pathway. Notably, phosphorylation of Plk1 at T210 (mouse numbering, T201 in *Xenopus*), which is catalyzed by Aurora A and is critical for Plk1 activation (*Macůrek et al., 2008*), as well as phosphorylation of Gwl at S442 (mouse numbering, 467 in *Xenopus*), increased in both datasets (*Figures 6B and 10A–B*). Similarly, both datasets consistently detected phosphorylation of Erk2 at T183 (mouse numbering, T188 in *Xenopus*), which reflects MEK-dependent activation of this kinase (*Payne et al., 1991*; *Figures 6B and 10A–B*). Among the conserved sites whose phosphorylation increases during meiosis, several are known targets of Cdk1 and Erk/MAPK (*Figure 10B*), consistent with the activation of these kinases during meiotic maturation. We experimentally confirmed the phosphorylation of Fak1 at S913, Erk2 at T188, and Plk1 at T201 by western blot (*Figure 10C*). Moreover, by microinjecting the Cdk1-specific inhibitor Cip1 in oocytes in order to prevent Cdk1 activation, we demonstrated that Cdk1 is required for phosphorylation of all these substrates, whether they are direct targets of Cdk1 or Erk/MAPK (*Figure 10C*). This aligns with previous reports indicating that MAPK pathway activation in oocytes occurs downstream of the initial activation of Cdk1 (*Santoni et al., 2024*). Interestingly, AKTS1/PRAS40 phosphorylation decreases during meiotic maturation in both species (*Figure 10A–B*). We experimentally validated the dephosphorylation of AKTS1/PRAS40 at S208 and showed that this event occurs at NEBD and depends on Cdk1 activity (*Figure 10C*). AKTS1/PRAS40 is a direct inhibitor of mTORC1 by competing with 4E-BP1 and S6K1 for binding to raptor (*Wang et al., 2007*; *Oshiro et al., 2007*). AKTS1/PRAS40 is phosphorylated on S184 (mouse numbering, S208 in *Xenopus*) by mTORC1, which leads to its dissociation from mTOR and enhances its ability to phosphorylate its substrates (*Oshiro et al., 2007*). Interestingly, previous studies suggest that the translation of ribosomal proteins, which are targets of mTOR signaling via the TOP motif located in their 5′-UTR, decreases during meiotic maturation (*Luong et al., 2020*). Our findings support these observations and provide additional evidence for the downregulation of mTOR activity upon meiotic resumption.

## Discussion

The amphibian oocyte and egg have long been a source of inspiration and a source of experimental opportunity for important areas of biology, such as nuclear organization, spindle formation, cell polarity, fertilization, mitosis and meiosis, endocrine signaling, and intracellular signal transduction. As experimental embryology gave way to genetic approaches in the late 20[th] century to study embryonic development, the attractiveness of amphibian oocytes declined.

The original advantages of the amphibian oocyte are now becoming more obvious, as it proves important to link complex cellular processes, like cell division, to signal transduction. These processes are difficult to study because they occur at a single cell or even at subcellular level. Indeed, many important cell-division-related processes involve transient changes occurring in specific compartments of the cell, like the endoplasmic reticulum, the mitotic apparatus, the nuclear and plasma membranes, or the cytoskeleton. Additionally, meiotic and mitotic divisions occupy a very short fraction of the life span of germ and somatic cells, making it hard to study them in cultured cells. Frog oocytes and eggs

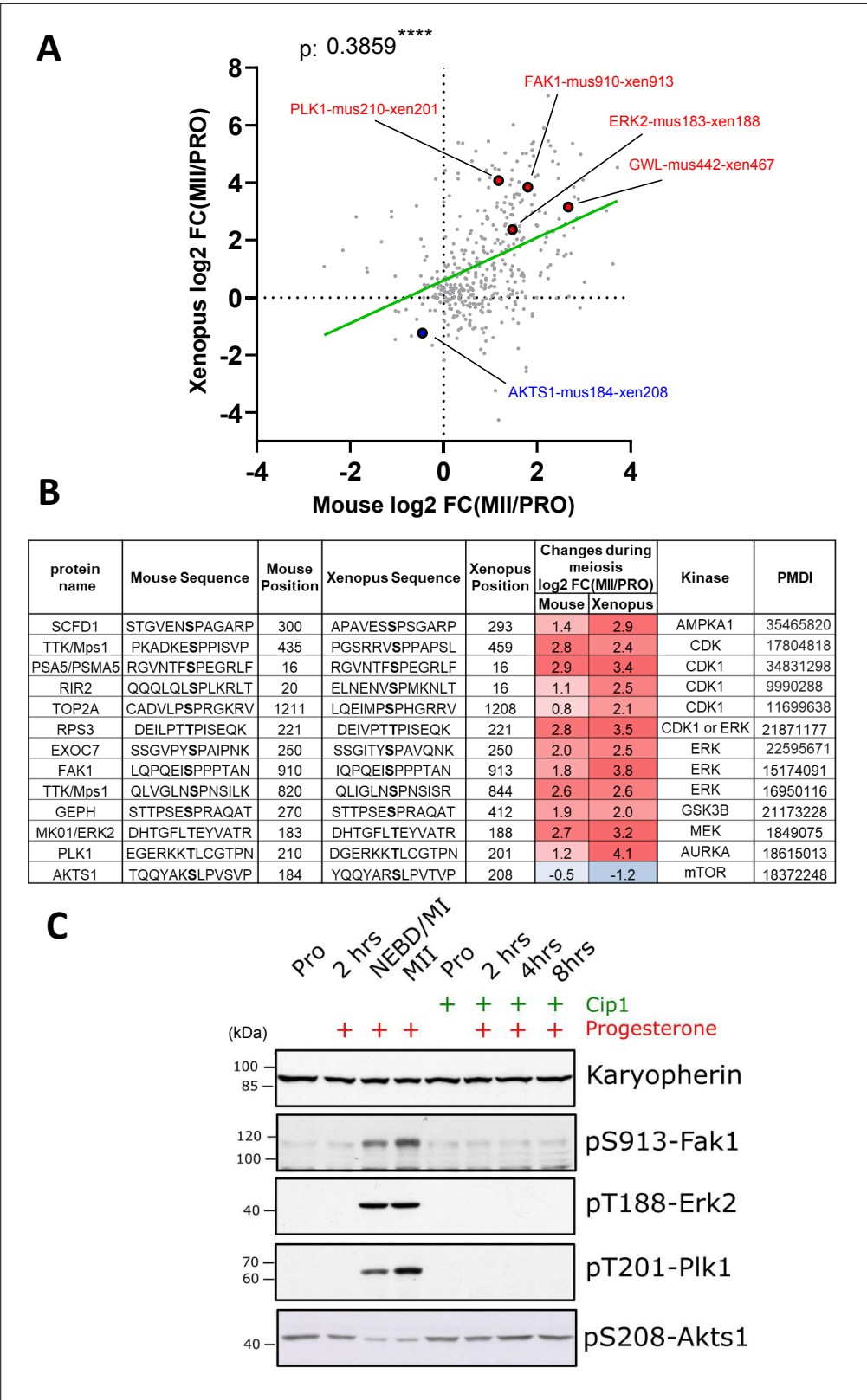

**Figure 10.** Comparison of phosphorylation changes during meiotic divisions in *Xenopus* and mouse oocytes. (**A**) Comparison of phosphorylation changes during meiotic divisions in *Xenopus* (our dataset) and mouse oocytes (***Sun et al., 2024***). Log2 fold changes in phosphorylation between prophase (PRO) and metaphase II are plotted. The Pearson correlation (*r*=0.39, p<0.0001) was calculated. (**B**) Selected examples of conserved phosphorylation

*Figure 10 continued on next page*

Figure 10 continued

sites between mouse and *Xenopus*. The kinase that was experimentally identified to phosphorylate the specific site is displayed together with the supporting reference. (**C**) Oocytes were microinjected with a specific inhibitor of Cdk1, Cip1. After overnight incubation, meiotic maturation was induced by progesterone. Oocytes were collected at different times after progesterone treatment. Karyopherin was used as a loading control. The phosphorylation of Fak1 at S913, Erk2 at T188, Plk1 at T201, and Akts1 at S208 was evaluated by western blot.

The online version of this article includes the following source data for figure 10:

**Source data 1.** Original files of the full raw uncropped blots of *Figure 10C*.

**Source data 2.** Original western blots indicating the relevant bands used in *Figure 10C*.

show exquisite synchrony of cell division, allowing to have large cohort of cells at the same stage of the cell cycle. Therefore, we and others have returned to the awesome power of the amphibian oocyte and egg to study the signal transduction, intracellular posttranslational changes surrounding the onset of meiotic and mitotic divisions, and fertilization at a single cell level.

Importantly, the ingenious and sensitive tools developed to study the transcriptional regulation are of little use in the study of a cell type that does not transcribe RNA and of events that are regulated post-translationally, such as meiotic and early embryonic divisions. For a long time, we lacked general tools for the identification and measurement of the activity of the regulators of these processes, impairing our ability to connect the morphological changes previously described by histology to the respective molecular events. One manner to investigate these pathways is using pharmacological inhibitors, but their non-specificity and the complexity of pathways pose serious limitations. For years, phosphorylation events have been extensively studied using candidate-based approaches, such as site-specific mutagenesis and phosphospecific antibodies. These targeted approaches have several caveats. The selection of the putative sites of phosphorylation is based on in silico evidence, limiting the analysis to proteins bearing canonical consensus sequences and hence our knowledge of novel phosphorylation motifs. When multiple putative phosphorylation sites are present in the same protein, mutagenesis of individual sites or combinations of sites becomes too complex, impairing identification of the phosphorylation sites used in vivo and their interdependency or redundancy. Moreover, the time-consuming production of phospho-specific antibodies is not scalable to high-throughput applications. Mass spectrometry has the potential of shedding light on these intricated networks. However, the current tools are unable to accurately quantify the level of proteins and phosphorylated proteins at the single cell level, except for the frog oocytes and eggs, thanks to their exceptional size and high protein content.

Oocyte meiotic maturation is a key biological process. First, it transforms the oocyte into a fertilizable cell at the origin of life of any animal. Second, since meiosis derives from mitosis, thanks to cellular adaptations that were necessary to produce a haploid gamete (preventing an S-phase between the two divisions, modifying the functioning of the spindle during meiosis I, establishing division arrests needed for the oocyte to grow or to wait for sperm, performing asymmetrical divisions to save nutrient reserves, etc.), its study provides the opportunity to understand how cell division can adapt and diversify in living systems, depending on specific cellular organization and specialized functions.

In this paper, we have applied the current state of the art of quantitative mass spectrometry to oocyte maturation and meiotic divisions using quantitative proteomics and quantitative phosphoproteomics. We have temporally correlated events with oocyte maturation induced naturally by progesterone in *Xenopus* full-grown oocytes and watched the progression through meiosis I to the natural arrest at the second meiotic metaphase. We have been able to correlate the proteomic measurements with the physiological changes of the oocyte, as it goes from an early prophase I state through to a second meiotic metaphase state. We find dramatic changes in the oocyte nucleus and the meiotic spindles and drastic modifications to the standard cell cycle. These dramatic changes are choreographed principally by signaling pathways employing kinases and phosphatases. In all animals, meiotic maturation relies on the dynamics of phosphorylation events, either during the signaling pathway that leads to Cdk1 activation or during the Cdk1 downstream period that orchestrates the structural events of the two meiotic divisions. The candidate-based approaches have provided a global scheme of this process based on the knowledge of a few kinases that have been extensively studied for years. However, this scheme still does not provide a very comprehensive picture of the meiotic process, since it does not take into account hundreds of unknown critical kinases, substrates, and phosphosites.

Our quantitative proteomics and phosphoproteomics approach allowed to obtain a much deeper and comprehensive dynamic picture of the phosphorylated sites during meiotic maturation. This has enabled us to describe and support with biochemical precision the various processes in the oocyte, determine the relative timing of various site phosphorylations and molecular processes, thereby challenging or suggesting the epistatic sequences of specific molecular pathways. Furthermore, we confirmed the importance of certain previously known phosphosites, identified the likely kinases or, in some cases, the kinase families responsible for substrate phosphorylation, and established connections between phosphorylation events and intracellular remodeling, as well as regulatory mechanisms such as translation and protein degradation.

In addition to explaining the order of diverse events of chromatin remodeling, nuclear organization, plasma membrane transformations, cytoskeletal changes of meiosis, our proteomic, and phospho-proteomic datasets should provide a large number of experimental avenues for exploring similar events in both mitotic and meiotic stages of diverse organisms. This molecular data will enable us to better understand regulatory pathways involving chromosome behavior, spindle formation, nuclear and plasma membrane regulation, and signal transduction, and derive insights far beyond the scope of meiosis and oocyte maturation. One such insight might come from a closer analysis of the quality control mechanisms in the oocyte. Both nucleopore remodeling (*King et al., 2019*) and E2/E3 ubiquitin ligases are involved in housekeeping and quality control, specifically in the removal of aggregated and orphan complex proteins, as the germ cell of an organism prepares to become the next generation.

## Methods

### Oocyte collection and in vitro maturation

The research with *Xenopus laevis* was performed under the oversight of the Harvard Medical Area Institutional Animal Care and Use Committee in accordance with HMS IACUC protocol IS00001365. Mature *Xenopus laevis* females were used after priming with pregnant mare serum gonadotropin (PMSG, 50 IU per frog). A sample of ovary tissue was surgically removed, rinsed in 1x Marc's modified Ringer's (MMR), and placed in OR2 buffer (83 mM NaCl, 2.5 mM KCl, 1 mM $CaCl_2$, 1 mM $MgCl_2$, 1 mM $Na_2HPO_4$, 5 mM HEPES, pH 7.8) and oocytes were manually defolliculated. Prophase oocytes were selected and kept in dishes with agar beds to avoid sticking to the dish. For the Cip1 experiment, prophase oocytes were microinjected with 60 ng of GST-XenCip1, as previously described (*Santoni et al., 2024*). Maturation was induced by incubation at 18°C with 10µg/mL progesterone in OR2 buffer.

Oocytes were flash frozen with liquid nitrogen in bunches of 10 or 50 for protein and phospho-proteomics measurements, respectively. All samples were taken in triplicates, each biological replicate was performed with different frogs. The first sample was taken from untreated oocytes at the time of progesterone treatment, the second timepoint 2 hr after progesterone treatment, with hourly time-points past that, up to 9 hr post-treatment.

### Oocyte lysis and digestion into peptides

For each condition, the oocytes were lysed at a ratio of 6µL buffer/embryo in the same buffer as (*Presler et al., 2017*) (0.25M sucrose, 1% replacement NP-40, 10mM EDTA, 25mM HEPES, 1mM TCEP, 10mM Combretastatin, 10µM Cytochalasin D, Roche Protease Inhibitor Mini EDTA free, and Phosstop inhibitor (Roche) at pH 7.2). The yolk was removed by spinning the samples at 4000g for 4min at 4°C. Both the supernatant and all lipids were transferred to a new tube. The concentration of HEPES was raised to 0.1M and SDS was added to 20%. After the addition of DTT to 5mM, the samples were denatured in a 60°C water bath for 20min. After the samples temperatures returned to room temperature, N-Ethylmaleimide (NEM) was added to 15mM, tubes were mixed thoroughly, and the reaction proceeded for 1hr. Then additional DTT was added to 5mM to quench the excess NEM. The samples were flash frozen and stored at –80°C until the methanol-chloroform precipitation (*Wessel and Flügge, 1984*). Protein was precipitated by methanol-chloroform precipitation. Due to the large volume of lysate (~1mL), the precipitation was performed in glass 15mL Corex tubes (No. 8441). The tubes were cleaned for a 2 hr chromic acid incubation and rinsed extensively in filtered water. Then they were autoclaved and before use for the precipitation, they were rinsed with HPLC $H_20$. The

centrifugations were done in a JA 50 30 Ti rotor in a Beckman Avanti J-30I centrifuge at 21,000g for either 6min or 15min. After the last supernatant was removed, the pellets were air dried for 15min and resuspended in 0.6mL of 6M guanidine, 50M EPPS pH 8.5 and transferred to a 2mL Eppendorf tube. The samples were incubated for 5min at 65°C and the protein amount was determined by a BCA Assay. The solution was diluted to 2M guanidine with 10mM EPPS pH 8.5. Lysyl endopeptidase (FUJI-FILM WAKO (129–02541) resuspended with HPLC H$_2$0 to ~2mg/mL) was added to a concentration of 20ng/mL and the initial digestion proceeded for 12hr at room temperature with gentle shaking. The samples were further diluted to 0.5M guanidine with 10mM EPPS pH 8.5 and Trypsin protease (Sequencing Grade Modified Trypsin, Promega) was added to a final concentration of 10ng/mL along with an additional 20ng/mL of LysC. The samples were incubated in a 37°C room for 8hr on a nutator.

### Phospho-peptide enrichment

To carry out the phosphoproteomic analysis, we followed the conventional steps of digesting the proteins from each time sample with trypsin and specifically labeling them with isobaric tags (TMT-10 reagents). Tagged samples and the phospho-peptides were enriched on an IMAC column. Importantly, we chose to multiplex peptides before the phospho-peptide enrichment to improve data quality. There is a trade-off of decreased yield and, therefore, depth, as it is not economical to label more than a few milligrams of material. We used 2.5–4 mgs of TMT-labeled peptides per replicate, enriched with 5μm Titanium Dioxide microspheres (GL Sciences 5020–75000) and fractioned as previously described (*Presler et al., 2017*). A typical yield of 50–80μg of peptides eluted from the column, with a median phospho-peptide enrichment of ~80%.

### Phosphatase treatment of *Xenopus* samples prepared for parallel phospho-enrichment

We adapted the method that we used to phosphatase-treat digested peptides (*Presler et al., 2017*) for the larger samples used here for the parallel phospho-enrichment. We chose to phosphatase treat when proteases were present in the samples (but after the protease digestion sequence described above) to enable the dephosphorylated peptides to have the cleavage pattern expected if they were present initially in the dephosphorylated state. We used the heat-labile Alkaline Shrimp Phosphatase (Affymetrix Inc, USA). We used a 3k Amicon Centrifugal Filter to exchange the enzyme storage buffer to 10mM EPPS pH 8.5 instead of Tris-HCl pH 7.5. We also concentrated the phosphatase to between 3–4U/μL (confirmed by p-nitrophenyl phosphate assay). We added an estimated 833 units of phosphatase to two of the samples built to contain representative mixture of peptides (TMT 131N, channel 10: 100 oocytes collected at the latest timepoint, and TMT 131 C, channel 11: 50 oocytes collected at the first timepoint combined with 50 oocytes collected at the last timepoint). To samples that were not phosphatase-treated, we added the same volume of blank phosphatase buffer: 5mM EPPS pH 8, 50% glycerol. To all samples, we added MgCl$_2$ to 10mM and EDTA to 0.1mM (to chelate metals that might compete with the magnesium necessary for enzymatic activity). All the samples were incubated with gentle shaking at room temperature for 12hr. The phosphatase was then inactivated by incubation with gentle shaking in an air incubator at 65°C for 15min. Peptide fractionation and TMT-MS3 LC/MS were carried out as done previously (*Presler et al., 2017*).

### Computing phospho-occupancy systematically

To compute phospho-occupancy, we used the phospho_occupancy_matlab package, available as part of our code repository (https://github.com/elizabeth-van-itallie/phospho_occupancy_matlab; *Van Itallie, 2024*) implementing the methods previously described in *Van Itallie et al., 2025*. We grouped phospho-peptides by unique phosphosite identifiers. In order to rigorously reflect the measurement accuracy and to take advantage of many measurements of the same phosphosite (including on peptides with different missed-cleavages or oxidized methionines), we used BACIQ (*Peshkin et al., 2019*) to compute the estimated trend with confidence intervals. Then, we created phosphosite sets to connect phosphosites on residues that could be on the same peptide. We made these assignments based on whether the individual phosphosites were ever measured with another phosphosite as a composite phosphosite. In order to determine occupancy, we needed to measure the phosphosite as a different form (usually not-phosphorylated). To do this, we identified non-phosphorylated peptides measured from the same experimental conditions that include the residue(s) of the phosphosite. Again, multiple

non-phosphorylated peptides can include the residues of interest and the same peptide sequences can be measured multiple times. Therefore, we again use BACIQ (*Peshkin et al., 2019*) to aggregate these measurements in a trend with confidence intervals.

## K-means clustering

The relative protein abundance data across subsequent hourly timepoints post-progesterone stimulation was clustered using the K-means clustering algorithm with cosine similarity distance. The number of clusters was selected to ensure proteins in each cluster are closely concordant and under-clustered for visualization purpose, resulting in 60 and 90 clusters for protein and phosphopeptides, respectively. The code to perform this clustering procedure is available as *Source code 1* and was run using MATLAB (version R2022b). The K-means clustering algorithm converges to local minima only, therefore, we performed multiple clustering realization to identify the most biologically plausible clusters, as shown in panels A-B. Resulting clusters are ordered by cardinality, most populated clusters first.

## Comparison of mouse and *Xenopus* phosphosites

To compare the dynamics of mouse and *Xenopus* phosphosites between two species, we used the *phospho_matching* package, available as part of our code repository (https://github.com/elizabeth-van-itallie/phospho_matching; *Van Itallie, 2025*) implementing the methods previously described (*Van Itallie et al., 2025*). We first matched the phosphosites using the protein sequence and phosphorylated residues, then filtered the list to only keep matched phosphosites measured in both datasets. For cross-species phosphomatching, we used the default parameters that are defined in the methods section in *Van Itallie et al., 2025*: BLOSUM90 substitution penalty matrix and blastp alignment results with E-value less than 1e-20.

## Mass spectrometry data mapping and analysis

Peptide-Spectra matches were performed as previously described (*Sonnett et al., 2018*). Assignment of MS2 spectra was performed using the SEQUEST (*Eng et al., 1994*) algorithm by searching the data against the appropriate proteome reference set acquired from Xenbase (*Xenopus laevis* assembly 9.1) along with common contaminants: human keratins and trypsin. The target-decoy strategy was used to construct a second database of reversed sequences that were used to estimate the false discovery rate on the peptide level. SEQUEST searches were performed using a 20 ppm precursor ion tolerance with the requirement that both N- and C-terminal peptide ends are consistent with the protease specificities of LysC and Trypsin. TMT (+229.162932 Da) was set as a static modification on N-termini and lysine peptides, and N-ethyl maleimide (+125.047679 Da) was set as a static modification on cysteine residues. Oxidation of methionine (+15.99492 Da) was set as a variable modification. A peptide level MS2 spectral assignment false discovery rate of 1% was obtained by applying the target-decoy strategy with linear discriminant analysis. Peptides of 7 amino acids length or longer were ranked by linear discriminant analysis score and were filtered to a ratio of 1% reverses/forwards +reverses. Peptides were assigned to proteins and a second filtering step to obtain a 1% FDR on the protein level was applied. Peptides that matched multiple proteins were assigned to the proteins with the most unique peptides. We only used Isolation Specificity >0.75 spectra.

## Estimation of the absolute abundance of proteins

The absolute protein concentration was estimated according to previously published methods (*Wühr et al., 2014*; *Peshkin et al., 2019*), based here on ion current prorated to the isobarically labeled fractions. The respective sample fractions were estimated using our previously published Bayesian approach (*Peshkin et al., 2019*), which integrates peptide signal and peptide-level measurement agreement into a maximum likelihood estimate of the true protein ratio and the associated confidence interval (*Peshkin et al., 2019*). See Appendix Supplementary Methods.

## Western blot and antibodies

Oocyte homogenization and SDS-PAGE were performed as previously described in *Santoni et al., 2024*. The following antibodies were used: Karyopherin (1:2000, goat, Santa Cruz sc-1863), pT188-pY190-Erk2 (1:2000, mouse, Cell Signaling 9106), pT201-Plk1 (1:1000, Abcam Ab39068), pS910-FAK (1:5000, rabbit, Invitrogen 4459 G) and pS183-Akts1/PRAS40 (1:5000, rabbit, Cell Signaling 5936).

The appropriate secondary HRP-coupled antibodies were used at 1:10,000 dilution (Jackson Immunoresearch).

## Acknowledgements

Marc W Kirschner and Catherine Jessus contributed equally to this work. This work was supported by the National Center for Scientific Research (CNRS) and Sorbonne University, the National Research Agency (ANR) grants 18-CE13-0013-01 (CJ) and ANR-23-CE12-0045-01 (EMD), Sorbonne University Emergence grant (EMD), the ARC foundation grant ARCPJA2023080006901 (EMD), Amaranth Foundation (LP), and the NIH OD award R24 OD031956 (LP and MWK).

## Additional information

### Funding

| Funder | Grant reference number | Author |
|---|---|---|
| Agence Nationale de la Recherche | 18-CE13-0013-01 | Catherine Jessus |
| Agence Nationale de la Recherche | 23-CE12-0045-01 | Enrico maria Daldello |
| Fondation ARC pour la Recherche sur le Cancer | ARCPJA2023080006901 | Enrico maria Daldello |
| Sorbonne Université | Emergence | Enrico maria Daldello |
| Amaranth Foundation | | Leonid Peshkin |
| National Institutes of Health | R24 OD031956 | Leonid Peshkin |

The funders had no role in study design, data collection and interpretation, or the decision to submit the work for publication.

### Author contributions

Leonid Peshkin, Conceptualization, Resources, Data curation, Software, Formal analysis, Supervision, Funding acquisition, Methodology, Writing – original draft, Writing – review and editing; Enrico maria Daldello, Formal analysis, Visualization, Writing – original draft, Writing – review and editing; Elizabeth S Van Itallie, Data curation, Software, Formal analysis, Writing – review and editing; Matthew Sonnett, Wilhelm Haas, Formal analysis, Methodology; Johannes Kreuzer, Data curation, Formal analysis; Marc W Kirschner, Conceptualization, Supervision, Methodology, Writing – original draft, Writing – review and editing; Catherine Jessus, Formal analysis, Supervision, Writing – original draft, Writing – review and editing

### Author ORCIDs

Leonid Peshkin https://orcid.org/0000-0002-6420-848X
Enrico maria Daldello https://orcid.org/0000-0002-0456-8950
Matthew Sonnett https://orcid.org/0000-0002-6795-1308
Johannes Kreuzer https://orcid.org/0000-0003-4716-2843
Marc W Kirschner https://orcid.org/0000-0001-6540-6130
Catherine Jessus https://orcid.org/0000-0001-9296-6336

### Ethics

The research with Xenopus laevis was performed under the oversight of the Harvard Medical Area Institutional Animal Care and Use Committee in accordance with HMS IACUC protocol IS00001365.

Reviewer #1 (Public review): https://doi.org/10.7554/eLife.104255.3.sa1
Reviewer #2 (Public review): https://doi.org/10.7554/eLife.104255.3.sa2
Reviewer #3 (Public review): https://doi.org/10.7554/eLife.104255.3.sa3
Author response https://doi.org/10.7554/eLife.104255.3.sa4

## Additional files

### Supplementary files

Supplementary file 1. Absolute concentrations of proteins.

Supplementary file 2. Changes in protein abundance during meiotic maturation.

Supplementary file 3. Changes in protein phosphorylation during meiotic maturation.

Supplementary file 4. List of proteins whose phospho-occupancy was calculated.

Supplementary file 5. Common phosphosites between *Xenopus* (this work) and mouse (*Sun et al., 2024*) datasets.

MDAR checklist

Source code 1. MATLAB code to perform k-means clustering of time series.

### Data availability

The mass spectrometry proteomics raw data and reference set of sequences have been deposited to the ProteomeXchange consortium via the PRIDE partner repository with the dataset identifier MassIVE MSV000094498. In addition, we developed a multi-functional Web portal to release the data and enable interactive interrogation (https://xenopus.hms.harvard.edu/index_oocyte.html). All of the aspects from overall protein trends to individual clusters to specific phosphosites and stoichiometry information are easily available. The portal provides per-protein link to external information about each human protein by symbol from GeneCards (https://www.genecards.org/) and respective *Xenopus* protein at Xenbase (https://www.xenbase.org/xenbase/).

The following previously published dataset was used:

| Author(s) | Year | Dataset title | Dataset URL | Database and Identifier |
|---|---|---|---|---|
| Peshkin L, Daldello EM, Itallie EV, Sonnett M, Kreuzer J, Haas W, Kirschner MW, Jessus C | 2025 | Decoding protein phosphorylation during oocyte meiotic divisions using phosphoproteomics | https://massive.ucsd.edu/ProteoSAFe/dataset.jsp?accession=MSV000094498 | MSV000094498, massive |

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

## Appendix 1

### Estimating the absolute abundance of proteins

Following previously published methods (*Wühr et al., 2014*; *Peshkin et al., 2015*), the absolute protein concentration was estimated here based on ion current (denoted $I$) prorated to the isobarically labeled fractions. The respective sample fractions were estimated using our published Bayesian approach (*Peshkin et al., 2019*), which integrates peptide signal and peptide-level measurement agreement into a maximum likelihood estimate of the true protein ratio and the associated confidence interval (BACIQ). Specifically, we want to construct the following relation applicable to all measured proteins:

$$log\left(C_S^P\right) = A_P \, log\left(Q_S^P\right) + B_P$$

where:

$Q_S^P$ is the isobarically labeled fractions of the total $I$ ion current;

$C_S^P$ is the absolute abundance (in Moles) of the protein $p$ at a stage $s$;

$N_P^T$ is the theoretical number of (unique) peptides obtained by 'in-silico digestion' script (available in the *Supplementary files 1–5*) using the following parameters: miss_cleav = 2, min_len = 7, max_len = 57, enz='[KR]' and FASTA file of the considered protein sequences;

$F_S^P$ is the fraction of the total signal observed in the isobaric peptide signal quantitation channels corresponding to the sample from stage $S$;

$I$ is the total ion current reported using only unique peptides as obtained from *cq_area_sum* column of export from the MS platform;

$A_P$ and $B_P$ are the linear regression parameters.

Using the calibration protein data, we can fit (using two-way regression to avoid any bias towards one of the variables) a linear function (1) in log/log space and extrapolate from it by estimating the regression parameters $A_P$ and $B_P$ universal for all proteins. To calibrate the protein measurements, we used previously published (*Peshkin et al., 2015*) concentrations $C_S^P$ for 5960 proteins in *Xenopus laevis* egg. The parameters $A_P$ and $B_P$ are obtained using symmetrical linear regression analysis (in log-log space; *Appendix 1—figure 1*) between $C_{egg}^P$ (concentration in the egg stage post-fertilization) and $Q_S^P$ (isobarically labeled fractions of the total $I$ ion current). Furthermore, we only used in the regression analysis proteins with highly confident mapping between proteins in both data sets by sequence homology. The highly confident mapping - reciprocal best hit (RBH) by sequence homology is defined by HMMER3 pipeline (*Savova et al., 2017*) with the perfect E-values both ways matches of proteins between data published in *Peshkin et al., 2015* and proteins in the current study. This filter leaves 617 proteins used in regression analysis as shown in *Appendix 1—figure 1* for one of three biological repeats (Clutch A).

Using the fitting parameters, we converted mass spectrometry measurements into absolute abundance for all proteins detected in the experiments for three biological repeats (Clutch A, B, and C). To additionally validate the identification and estimated concentrations of proteins in this study, we performed the comparison with previously published data by *Wühr et al., 2014* matching proteins via respective human gene symbol as assigned via reciprocal HMMER3 run. *Appendix 1—figure 2* shows a Venn diagram illustrating that vast majority of proteins we identified (5063 or 86%) were also detected and confirmed in previously published data, while there are 800 novel proteins. Our current study was done for quantitative comparison across multiple time points and thus, being a labeled mass spectrometry experiment, could not be as deep as the unlabeled single sample unlabeled mass spectrometry study by *Wühr et al., 2014*. The 5063 proteins common between our data and (*Wühr et al., 2014*) are used to make comparison of proteins concentrations as shown in *Appendix 1—figure 3*. Pearson's correlation between previously published protein concentrations and obtained oocyte proteins in this study is 0.81. The maximum deviation of the extrapolated value from previously estimated value is ca. 3 nM. The differences between studies might result from the biological variation between frogs, differences in gene symbol assignment between studies or differences in the sample preparation.

### Identification and exclusion of follicular cells proteins

Oocytes are surrounded in ovary by an envelope of follicular cells, blood vessels, and connective tissue. To study the dynamics of protein expression in oocytes after progesterone stimulation, we

physically removed the follicular matter with forceps. This process is imperfect, leaving some amount of follicular matter attached to the oocytes. To control for admixture of the follicular layers, the removed follicular material was separately collected and profiled as one of the samples in mass spectrometry. The average relative protein abundance in oocyte was compared to follicular protein expression as shown in *Appendix 1—figure 4* across a total of 7062 proteins. *Appendix 1—figure 4* shows oocyte- specific genes FETUB and H1FOO, as well as the follicular cells-specific proteins HBG1 and TUBB6. The cut-off for selecting 'exclusively follicular proteins' was somewhat arbitrarily selected at 0.6 orders difference in log-log space or roughly fourfold enrichment in the follicle taking into account that oocyte material is impure and itself contains substantial follicular residuals. These over-expressed 648 follicular proteins were removed from the oocyte protein analysis since the dynamics of these proteins reflects the artifactual fluctuation in the the presence of the remaining follicular cells in oocyte samples.

## Occupancy calculation

*Appendix 1—figure 5* illustrates the occupancy calculation for a single peptide where four phospho-forms are measured.

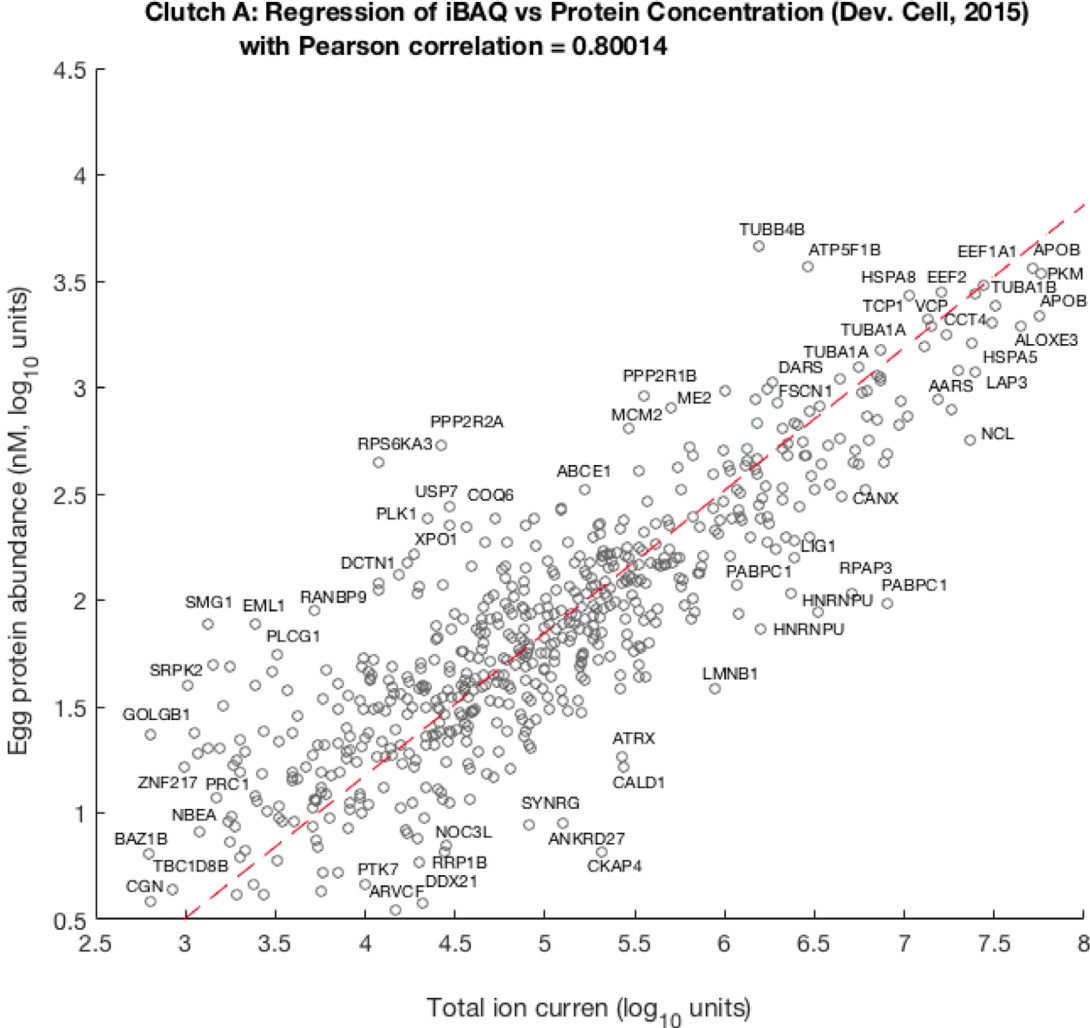

**Appendix 1—figure 1.** Scatter plot of per-channel pro-rated ion current $Q_S^P$ (isobarically labeled fractions of the total $I$ ion current) adjusted by theoretical number of tryptic peptides against egg protein abundance as previously published by *Peshkin et al., 2015*. Pearson correlation between these quantities is ca. 0.8, the best linear fit leads to the following relation: $\log\left(C_S^P\right) = 0.67*\log\left(Q_S^P\right) - 1.5$

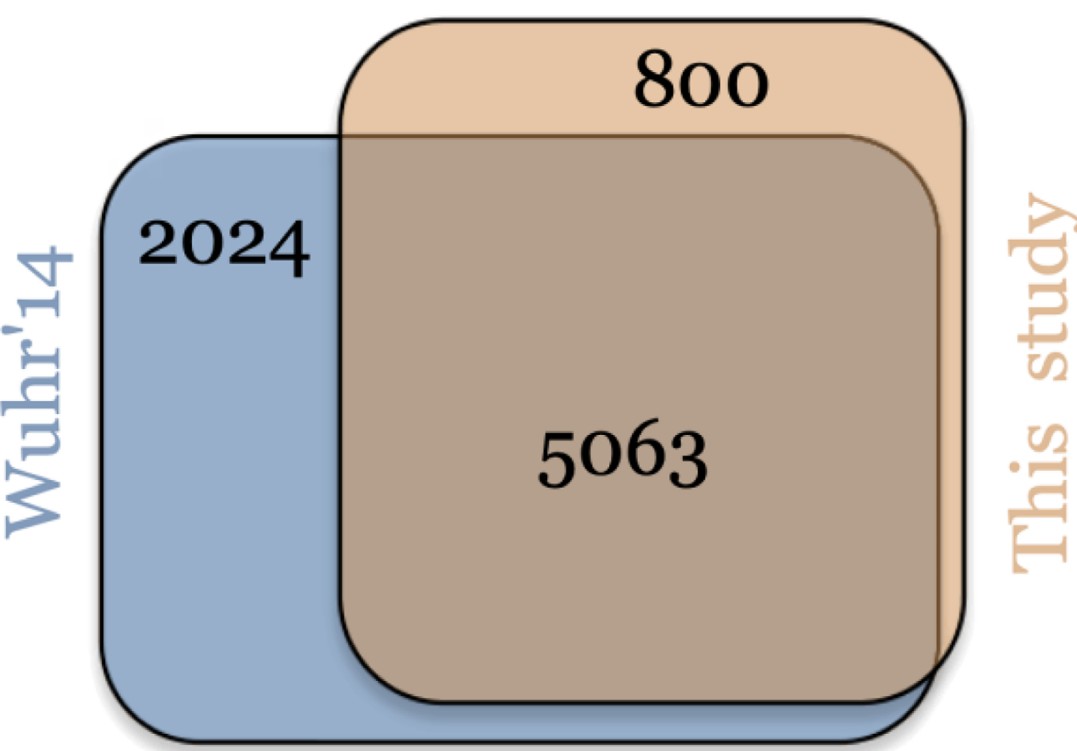

**Appendix 1—figure 2.** Venn diagram comparing proteins identified in this study to previously published by *Wühr et al., 2014*. This study has 800 unique proteins versus 2024 unique proteins in the data by *Wühr et al., 2014*. There are 5063 proteins in common.

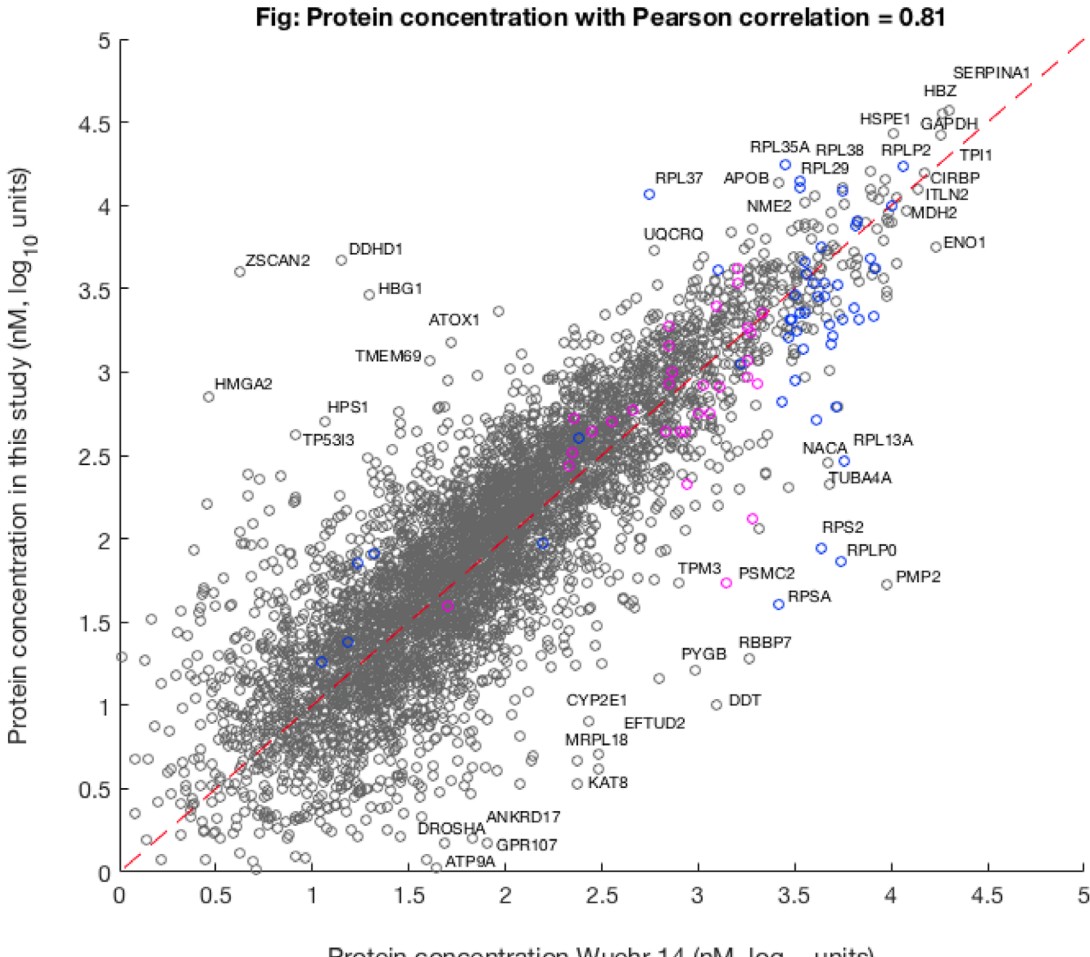

**Appendix 1—figure 3.** Comparison of average concentration (i.e. mean value of three different oocyte samples) with egg data published previously by *Wühr et al., 2014*. The magenta points show the location of proteasome units and the blue points show the ribosomal proteins.

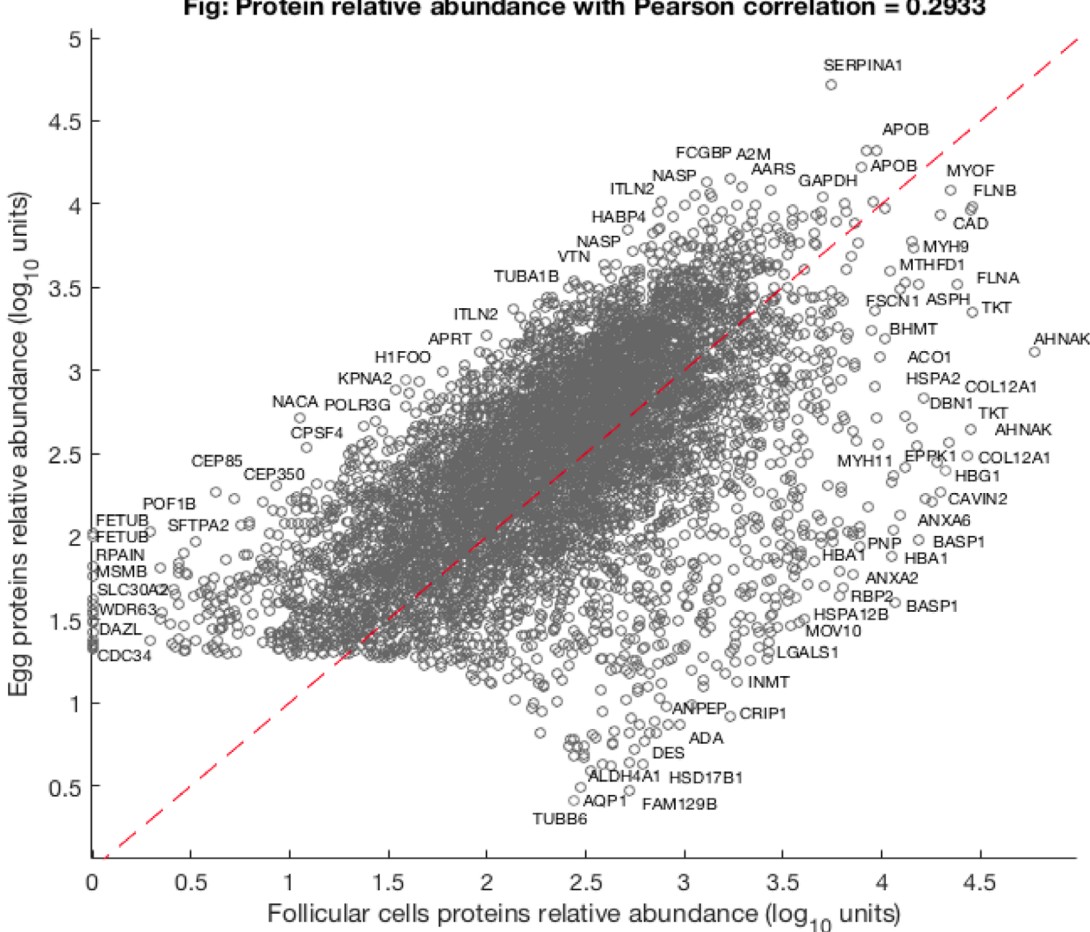

**Appendix 1—figure 4.** A scatter plot of a relative protein abundance in oocyte against abundance in follicular tissue. The low abundant proteins are naturally masked in the follicular sample as they are contrasted to 10 oocyte samples, thus severely underrepresented in a pooled sample.

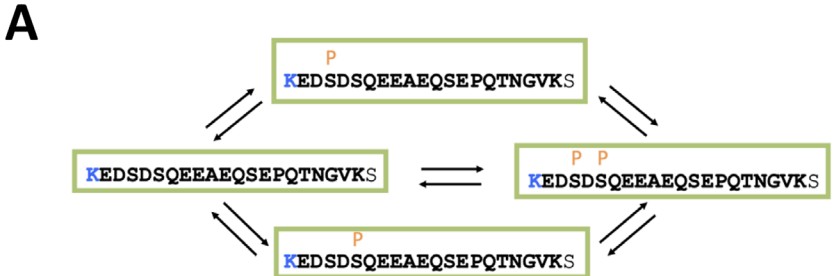

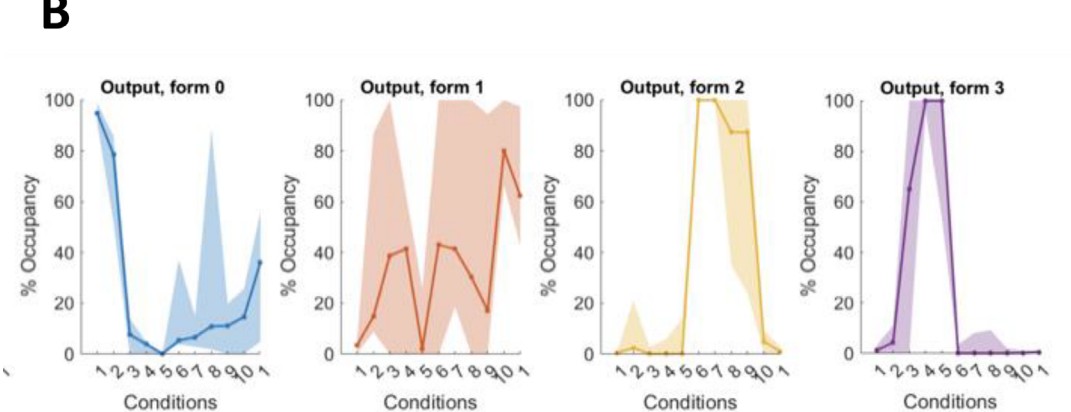

**Appendix 1—figure 5.** An illustration of occupancy calculation for a single peptide where four phospho-forms are measured. The dephosphorylated, singly phosphorylated on a serine in one of two alternative positions, and doubly phosphorylated on both serines. All interconversions (**A**) across four forms are allowed by our calculation. The resulting stoichiometry and respective confidence intervals (**B**) estimated for each form following *Presler et al., 2017* and BACIQ *Peshkin et al., 2019*. Points 1–9 are the time points post-progesterone stimulation, the two rightmost points are protease-treated samples.

