## [Editor Report · eLife Assessment]

This **important** paper describes a comprehensive quantitative phospho-proteomic analysis of the meiotic progression of *Xenopus* oocytes. Using time-resolved proteomic analyses, the authors provide insights into changes in protein levels and phosphorylation states to an unprecedented depth, quality, and quantitative detail. The key findings are **compelling** and offer a helpful resource for the scientific community.

---

## [Referee Report · Reviewer #1 (Public review)]

In the revised version of the manuscript, the authors have adequately addressed all our concerns. The authors should spell check their manuscript, e.g., correct phosphor-site to phospho-site, etc.

Summary:

The study aims to create a comprehensive repository about the changes in protein abundance and their modification during oocyte maturation in *Xenopus laevis*.

---

## [Referee Report · Reviewer #2 (Public review)]

Summary:

The authors analyzed Xenopus oocytes at different stages of meiosis using quantitative phosphoproteomics. Their advanced methods and analyses revealed changes in protein abundances and phosphorylation states to an unprecedented depth and quantitative detail. In the manuscript, they provide an excellent interpretation of these findings, putting them in the context of past literature in *Xenopus* as well as in other model systems. The clarity of these explanations improved significantly in the revised version of the manuscript, and several minor imprecisions have been corrected as well.

Strengths:

High-quality data, careful and detailed analysis, and outstanding interpretation in the context of the large body of literature.

Weaknesses:

Merely a resource, none of the findings are tested in functional experiments.

I am very impressed by the quality of the data and the careful and detailed interpretation of the findings. In this form, the manuscript will be an excellent resource to the cell division community in general, and it presents a very large number of hypotheses that can be tested in future experiments. *Xenopus* has been and still is a popular and powerful model system that led to critical discoveries around countless cellular processes, including the spindle, nuclear envelope, and translational regulation, just to name a few. This also includes a huge body of literature on the cell cycle describing its phosphoregulation. It is indeed somewhat frustrating to see that these earlier studies using phospho-mutants and phospho-antibodies were just scratching the surface. The phosphoproteomics analysis presented here reveals much more extensive and much more dynamic changes in phosphorylation states. Thereby, in my opinion, this manuscript opens a completely new chapter in this line of research, setting the stage for more systematic future studies.

---

## [Referee Report · Reviewer #3 (Public review)]

Summary:

The authors performed time-resolved proteomics and phospho-proteomics in *Xenopus* oocytes from prophase I through the MII arrest of the unfertilized egg. The data contains protein abundance and phosphorylation sites of a large number set of proteins at different stages of oocyte maturation. The large sets of data are of high quality. In addition, the authors discussed several key pathways critical for the maturation. The data is very useful for researchers, not only researchers in *Xenopus* oocytes but also those in oocyte biology in other organisms.

Strengths:

The data of proteomics and phospho-proteomics in *Xenopus* oocyte maturation is very useful for future studies to understand molecular networks in oocyte maturation.

Weaknesses:

Although the authors offered molecular pathways of the phosphorylation in translation, protein degradation, cell cycle regulation, and chromosome segregation. The authors did not check the validity of the molecular pathways based on their proteomic data by experimentation. But this is not essential since this is a resource paper.

---

## [Author Response]

The following is the authors’ response to the original reviews

**Public Reviews:**

**Reviewer #1 (Public review):**
Summary:The study aims to create a comprehensive repository about the changes in protein abundance and their modification during oocyte maturation in *Xenopus laevis*.Strengths:The results contribute meaningfully to the field.Weaknesses:The manuscript could have benefitted from more comprehensive analyses and clearer writing. Nonetheless, the key findings are robust and offer a valuable resource for the scientific community.

We would like to thank the reviewer for his/her positive feedback on our article. The public review points out that ‘The manuscript could have benefited from more comprehensive analyses and clearer writing.’ We have rewritten several sections and provided more detailed explanations of the analysis and interpretation of some data (see below for details). We have also followed all of the reviewer's recommendations, some of which specifically highlighted areas lacking clarity. We would also like to thank the reviewer for pointing out some errors, for which we apologize, and which have now been corrected. We sincerely appreciate the reviewer's thorough work, as it has greatly enhanced the clarity and precision of the manuscript.

**Reviewer #2 (Public review):**
Summary:The authors analyzed *Xenopus* oocytes at different stages of meiosis using quantitative phosphoproteomics. Their advanced methods and analyses revealed changes in protein abundances and phosphorylation states to an unprecedented depth and quantitative detail. In the manuscript they provide an excellent interpretation of these findings putting them in the context of past literature in *Xenopus* as well as in other model systems.Strengths:High quality data, careful and detailed analysis, outstanding interpretation in the context of the large body of the literature.Weaknesses:Merely a resource, none of the findings are tested in functional experiments.I am very impressed by the quality of the data and the careful and detailed interpretation of the findings. In this form the manuscript will be an excellent resource to the cell division community in general, and it presents a very large number of hypotheses that can be tested in future experiments. *Xenopus* has been and still is a popular and powerful model system that led to critical discoveries around countless cellular processes, including the spindle, nuclear envelope, translational regulation, just to name a few. This also includes a huge body of literature on the cell cycle describing its phosphoregulation. It is indeed somewhat frustrating to see that these earlier studies using phosphomutants and phospho-antibodies were just scratching the surface. The phosphoproteomics analysis presented here reveals much more extensive and much more dynamic changes in phosphorylation states. Thereby, in my opinion, this manuscript opens a completely new chapter in this line of research, setting the stage for more systematic future studies.

We thank the reviewer for his/her extremely positive comments. The public review points out that ‘none of the findings are tested in functional experiments.’ This is entirely accurate. We focused our work on obtaining the highest quality proteomic and phosphoproteomic data possible, and then sought to highlight these data by connecting them with existing functional data from the literature. This approach has opened up research avenues with enormous, previously unforeseen potential, in a wide range of biological fields (cell cycle, meiosis, oogenesis, embryonic development, cell biology, cellular physiology, signaling, evolution, etc.). We chose not to delay publication by experimentally investigating the narrow area in which we are specialists (meiotic maturation), while our data offer a vast array of research opportunities across various fields. Our goal was, therefore, to present this extensive dataset as a resource for different scientific communities, who can explore their specific biological questions using our data. This is why we submitted our article to the ‘Repository’ section of eLife. Nevertheless, in the context of the comparative analysis of the mouse and *Xenopus* phosphoproteomes performed at the reviewer’s request, we felt it was important to complement this new section with functional experiments that not only validate the proteomic data but also provide new insights into certain proteins and their regulation by Cdk1 (new paragraph lines 824-860 and new Figure 9).

We are also grateful to the reviewer for the recommendation to improve the manuscript by including more comparisons between our *Xenopus* data and those from other systems. We have followed this suggestion (see below), which has significantly enriched the article (new paragraph lines 824-860 and new Figure 9).

**Reviewer #3 (Public review):**
Summary:The authors performed time-resolved proteomics and phospho-proteomics in *Xenopus* oocytes from prophase I through the MII arrest of the unfertilized egg. The data contains protein abundance and phosphorylation sites of a large number set of proteins at different stages of oocyte maturation. The large sets of the data are of high quality. In addition, the authors discussed several key pathways critical for the maturation. The data is very useful for the researchers not only researchers in *Xenopus* oocytes but also those in oocyte biology in other organisms.Strengths:The data of proteomics and phospho-proteomics in *Xenopus* oocyte maturation is very useful for future studies to understand molecular networks in oocyte maturation.Weaknesses:Although the authors offered molecular pathways of the phosphorylation in the translation, protein degradation, cell cycle regulation, and chromosome segregation. The author did not check the validity of the molecular pathways based on their proteomic data by the experimentation.

We thank the reviewer for his/her positive comments. The public review points out that ‘The author did not check the validity of the molecular pathways based on their proteomic data by the experimentation.’ This is entirely accurate. We focused our work on obtaining the highest quality proteomic and phosphoproteomic data possible, and then sought to highlight these data by connecting them with existing functional data from the literature. This approach has opened up research avenues with enormous, previously unforeseen potential, in a wide range of biological fields (cell cycle, meiosis, oogenesis, embryonic development, cell biology, cellular physiology, signaling, evolution, etc.). We chose not to delay publication by experimentally investigating the very narrow area in which we are specialists (meiotic maturation), while our data offer a vast array of research opportunities across various fields. Our goal was, therefore, to present this extensive dataset as a resource for different scientific communities, who can explore their specific biological questions using our data. This is why we submitted our article to the ‘Repository’ section of eLife. Nevertheless, in the context of the comparative analysis of the mouse and *Xenopus* phosphoproteomes performed at the reviewer’s request, we felt it was important to complement this new section with functional experiments that not only validate the proteomic data but also provide new insights into certain proteins and their regulation by Cdk1 (new paragraph lines 824-860 and new Figure 9).

We have also followed all of the reviewer's recommendations and thank him/her, as the suggestions have significantly enhanced the manuscript.

**Recommendations for the authors:**

**Reviewer #1 (Recommendations for the authors):**
(1) Fig. 1 -> In the Figure legend ‘mPRβ’ is called ‘mPRb’ In the Figure, it is indicated that PKA substrates are always activated by the phosphorylation. As the relevant substrates and the mode-of-action of the Arpp19 phosphorylation are not clear at the moment, this seems to be preliminary. It could for example also be conceivable that PKA phosphorylation inhibits a translation activator. In addition, the PG-dependent translation of RINGO/Speedy should be included in the model.

We fully agree with the reviewer. PKA substrates can either be activators of the Cdk1 activation pathway, which are inhibited by phosphorylation by PKA, or repressors of the same pathway, which are activated by phosphorylation by PKA. This is now illustrated in the new Fig. 1. In addition, we have also included RINGO/Speedy in the model and in the text (lines 78-79) and corrected ‘mPRb’ in the legend.

(2) Lane 51-52 -> it is questionable if the meiotic divisions can be called ‘embryonic processes’

We agree with the reviewer comment, and we have removed the word ‘embryonic.’

(3) Lane 53 and lane 106-107 -> recent data have indicated that transcription already starts during cell cycle 12 and 13 in most cells (e.g. Blitz and Cho: Control of zygotic genome activation in *Xenopus* (2021))

We apologize for this mistake. The text has been corrected and the reference added (lines 53 and 107).

(4) Lane 61-62 -> ‘MI’ and ‘MII’ are given as abbreviation for ‘first and second meiotic spindle’

The text has been clarified to explain that MI is referred to metaphase I and MII stands for metaphase II (lines 61-64).

(%) Lane 131-132 -> ‘single-cell’ is mentioned redundantly in this sentence.

The sentence has been corrected (lines 131-132).

(6) Fig. 2B -> it is not explained what is plotted as ‘Average levels’ on the x-axis. Is it the average of expression over all samples or at a given time point? Are the values given as a concentration or are the values normalized? If so, how were they normalized?

We agree with the reviewer comment that ‘Average levels’ may have been unclear. In the new Fig. 2B, we have re-plotted the graph using the average protein concentration during meiosis, measured as described in the Methods section.

(7) In Fig. 2-supplement 3E -> from the descriptions it is not entirely clear to me what the difference to the data in Fig. 2B is?

We thank the reviewer for his/her question regarding the relationship between the data in Fig. 2B and Fig. 2-supplement 3E. We confirm that the raw data visualized in Fig. 2-supplement 3E are the same as those in Fig. 2B. However, in Fig. 2-supplement 3E, the data are color-coded differently to highlight the number of proteins whose concentrations change during meiotic divisions, based on the threshold adopted. The legend of Fig. 2-supplement 3E has been modified to clarify this point.

(8) Lane 225-226 -> Kifc1 is a minus-end directed motor

This mistake has been corrected (lines 232-233).

(9) Lane 271 -> Serbp1, here mentioned to be involved in stabilization of mRNAs, has also been implicated in the regulation of ribosomes (e.g. Leesch et al. 2023). Regarding the overall topic of this manuscript, this could be mentioned as well.

We agree with the referee that the important role of Serbp1 in the control of ribosome hibernation needs to be mentioned. We have included this point in the revised manuscript together with the reference (lines 277-279).

(10) Lane 360-363 -> it is mentioned that APPL1 and Akt2 act ‘to induce meiosis.’ Furthermore, in the Nader et al. 2020 paper, Akt2 phosphorylation is reported to happen within 30min after PG treatment. In the present work, they only seem to get phosphorylated when Cdk1 is activated. Is there an explanation for this discrepancy?

Indeed, Nader et al. (2020) indicate that Akt2 is phosphorylated on Ser473 (actually, they should have mentioned Ser474, which is the phosphorylated residue on Akt2; Ser473 corresponds to the numbering of Akt1) between 5 and 30 minutes post-Pg, which supports their hypothesis of an early role for this kinase. However, these conclusions should be taken with caution, considering that their functional experiment using antisense against Akt2 depletes only 25% of the protein, the antibody used to visualize Akt2 phosphorylation also recognizes phosphorylated Akt1 and Akt3, and they did not analyze phosphorylation of the protein after 30 minutes. Therefore, we cannot determine whether the level observed at 30 minutes represents a maximum or if it is just the onset of the phosphorylation that peaks later, possibly after activation of Cdk1, for example.

Regarding our measurements: we clearly observe phosphorylation of Akt2 following Cdk1 activation on Ser131. We did not detect Akt2 phosphorylation on Ser474, but since our measurements started 1 hour post-Pg, this protein may have returned to a dephosphorylated state on Ser474.

Therefore, the observations of Nader et al. and ours involve different residues and different phosphorylation kinetics, Nader et al. limiting their analysis to the first 30 minutes, whereas we started at 1 hour.

We have revised the manuscript text to make these aspects clearer (lines 387-392).

(11) Fig. 3B -> it could be made clearer in the Figure that all these sites belong to class I

A title “Class I proteins” has been added in Fig. 3B to clarify it.

(12) Lane 433-434 -> the authors write that the proteomic data of this study confirm that PATL1 is accumulating during meiotic maturation. However, in Fig. 2B PATL1 is not among the significantly enriched proteins.

We apologize for this error. Indeed, PATL1 protein is not significantly enriched. The text has been corrected (lines 461-465).

(13) Fig. 4B -> Zar2 is color-coded to increase in abundance. This is clearly different to published results and what is shown in Figure 2B of this manuscript.

Indeed, our dataset shows that the quantity of Zar2 decreases. This does not appear anymore in Figure 2B since Zar2 average concentration cannot be estimated. We made an error in the color coding, which has now been corrected in Figure 4B.

(14) Lane 442-444 -> it might be worth mentioning that the interaction between CPEB1 and Maskin, and thus probably its role in regulation of translation, could not be reproduced in other studies (Minshall et al.: CPEB interacts with an ovary-specific eIF4E and 4E-T in early *Xenopus* oocytes (2007) or Duran-Arque et al.: Comparative analyses of vertebrate CPEB proteins define two subfamilies with coordinated yet distinct functions in post-transcriptional gene regulation (2022)).

This clarification is now mentioned in the text, supported by the two references that have been added (lines 471-477).

(15) Lane 483-485 -> The meaning of these sentences is not entirely clear to me. What exactly is the similarity with the function of Emi1? What does ‘...binding of Cyclin B1...’ mean (binding to which other protein?). What is the similarity between Emi1 and CPEB1/BTG4, both of which are regulators of mRNA stability/polyadenylation?

We apologize if these sentences were unclear. Our intention was to emphasize the central role of ubiquitin ligases in regulating multiple events during meiotic divisions. We used SCF^βTrCP^, a well-studied ubiquitin ligase in *Xenopus* and mouse oocytes during meiosis, as an example. SCF^βTrCP^ regulates the degradation of several substrates, including Emi1, Emi2, CPEB1, and Btg4, whose degradation or stabilization is essential for the proper progression of meiosis. Lastly, we highlighted that these regulatory processes, mediated by protein degradation, may be conserved in mitosis, as for example the destruction of Emi1. We have rewritten this paragraph for clarity (lines 513-518).

(16) Lane 521-522 and 572-573 -> the authors write that Myt1 was not detected in their proteome. However, in Fig. 6A they list ‘pkmyt1’ as a class II protein. On Xenbase, ‘pkmyt1’ is the Cdk1 kinase, ‘Myt1’ is a transcription factor, so the authors might have been looking for the wrong protein.

We thank the reviewer for this accurate observation. We have modified the text to correct this error (lines 554 and 607).

(17) Lane 564-565 -> The authors state that Cdk1 activity can be measured by analyzing Cdc27 S428 phosphorylation. However, in vivo the net phosphorylation of a site is always depending on the relevant kinase and phosphatase activities. As S428 is a Cdk1 site, it is not unlikely that it is dephosphorylated by PP2A-B55, which by itself is under the control of Cdk1. Do the authors have direct evidence that the change in phosphorylation of S428 can only be attributed to the changes in Cdk1 activity?

There is evidence in the literature that Cdc27 is dephosphorylated by PP2A (Torres et al., 2010). In *Xenopus* oocytes, PP2A activity is high during prophase (Lemonnier et al., 2021) and decreases at the time of Cdk1 activation, mediated by the Greatwall-ENSA/Arpp19 system, remaining low until MII (Labbé et al., 2021). Therefore, the period where fluctuations in Cdk1 activity are difficult to assess, from NEBD to MII, corresponds to a phase of inhibited PP2A activity. As a result, the phosphorylation level of Cdc27 reflects primarily the activity of Cdk1. We have added this clarification in the text (lines 597-600).

(18) Fig. 7C and 7D -> in 7C, for Nup35/Nup53 there is a phospho-peptide GIMEVRS(60)PPLHSGG. In Fig. 7D phosphorylation of GVMEMRS(59)PLFSGG is analyzed. Is this the same phosphosite/region of Nup35/Nup53? How can there be a slightly different version of the same peptide in one protein? Are these the L- and S-version of Nup35/Nup53? It is also very surprising that the two phosphosites belong to different classes, class III and class II, respectively.

We thank the reviewer for this observation. The peptides GIMEVRS(60)PPLHSGG and GVMEMRS(59)PLFSGG correspond to the same phosphorylation site in the L and S versions of *Xenopus laevis* Nup35, respectively. The L version peptide was classified as Class III, while the S version was not assigned to any class due to its high phosphorylation level in prophase, which prevented it from meeting the log_2_ fold-change threshold of 1 required by our analysis to detect significant differences.

(19) Table 1 -> second last column is headed ‘Whur, 2014’

The typo has been corrected.

(20) Fig. 8 -> Why are all the traces starting at t=1h after PG?

The labeling of the graphs in Fig. 8 has been corrected, and the traces now begin at t0.

(21) Lane 754 -> Although a minority, there are also some minus-end directed kinesins, e.g. Kifc1

We agree with the reviewer. We should have mentioned that, in addition to dyneins, some kinesins are minus-end directed motors, especially since one of them, Kifc1, is regulated at the level of its accumulation. We have rephrased the relevant sentences to incorporate this observation (lines 790-793).

(22) Section ‘Assembly of microtubule spindles and microtubule dynamics’ -> Although this section clearly has a strong focus on phosphorylation, it might be worth mentioning again that many regulators of the microtubule spindle, e.g. TXP2, are among the upregulated proteins in Fig. 2B/C

We have already discussed that the protein levels of certain key regulators of the mitotic spindle (Tpx2, PRC1, SSX2IP, Kif11/Eg5 among others) are subject to control during meiotic maturation in a previous chapter ‘Protein accumulation: the machinery of cell division and DNA replication’ (lines 230-239). We agree with the reviewer that this important observation can be mentioned again at the beginning of this chapter on phosphorylation control. We have added a sentence regarding this at the start of the paragraph (lines 774-775).

**Reviewer #2 (Recommendations for the authors):**
While I find the manuscript excellent and detailed already in its current form, I would appreciate including even more comparisons to other systems. In particular, a similar phosphoproteomics experiment has been performed in starfish oocytes undergoing meiosis (Swartz et al, eLife, 2021), and there are several studies on mitosis of diverse mammalian cells. It would be very exciting to see to what extent changes are conserved.

We thank the reviewer for this recommendation, which we have attempted to follow. We have matched our dataset of mass spectrometry using the the phosphor-occupancy_matlab package, available as part of our code repository (https://github.com/elizabeth-van-itallie) previously described in (Van Itallie et al, 2025). Unfortunately, we were unable to match our dataset with the data from Swartz et al. (2021) on starfish oocyte due to the low sequence conservation. However, we have compared our dataset with the dataset from Sun et al. (2024) on mouse oocyte maturation. We identified a total of 408 conserved phosphorylation sites, which mapped to 320 proteins in *Xenopus* and 277 in mice (refer to a new paragraph: lines 824-860, new Figure 9, Methods: lines 1011-1032 and 1060-1065, and Appendix 7). The phosphorylation patterns during meiosis showed a significant cross-species correlation (Pearson r = 0.39, p < 0.0001; see new Figure 9A), demonstrating the evolutionary conservation of phosphoproteomic regulation. Important phosphorylation events, including Plk1 at T201, Gwl at S467, and Erk2 at T188, were upregulated in both species, in line with the activation of the Cdk1 and MAPK signaling cascades (Figure 6B, new Figure 9A-B). We validated several of these phosphorylation sites by western blotting and demonstrated their dependency on Cdk1 activation (new Figure 9C). Together, these findings reinforce the notion that fundamental phospho-regulatory pathways are conserved during oocyte maturation in vertebrates.

**Reviewer #3 (Recommendations for the authors):**
(1) Page 6, the first paragraph of Results section: Please describe the method on how the authors measured and quantified the proteomes in different stages of *Xenopus* oocyte maturation briefly. Without the experimental design, it is very hard to evaluate the results in the following paragraphs.

As requested by the reviewer, we added a few sentences describing the method of proteomics and phosphoproteomics measurements in oocytes resuming meiosis (lines 151-158).

(2) In the phospho-proteome, it is better to classify the amino acids for the phosphorylation such as Ser, Thr, and Tyr. Particularly how many tyrosine phosphorylations are in the list.

Our phosphosites dataset contains 80% Ser, 19.9% Thr, and 0.01% Tyr. Phospho-Tyr is slightly less abundant than what has been described in the literature in most cells ‘roughly 85-90% of protein phosphorylation happens on Ser, ~10% on Thr, and less than 0.05% on Tyr’ after Sharma et al., 2014. The same observation was made regarding the distribution of phosphorylated amino acids in mouse oocytes, where phospho-Tyr abundance is relatively diminished in oocytes compared to mouse organs (Sun et al., 2024). These observations are now reported in the manuscript (lines 309-313).

(3) In class II (Figure 3), when Cdk1 (line 326) is a major kinase, how many phosphorylation sites are a target of Cdk1 (with the Cdk1-motif)? Moreover, do the authors find any other consensus sequences for the phosphorylation? Those are either known or unknown. This information would be useful for the readers.

We thank the reviewer for this valuable comment. To address it, we used the kinase prediction server (https://kinase-library.phosphosite.org/kinase-library/score-site) to analyze Class II phosphosites. These new results are mentioned in lines 340-349 and illustrated in a new Figure (Figure 3—figure supplement 1A). We identified 303 sites predicted to be phosphorylated by Cdk1. Of these, 166 were also predicted as Erk1/2 targets, reflecting the similarity between Cdk1 and Erk1/2 consensus motifs.

Cdk1 substrate phosphorylation is governed by more than just the presence of a consensus sequence. In addition to its preference for the (S/T)P×(K/R) motif, Cdk1/cyclin complexes achieve specificity through docking interactions with short linear motifs (SLiMs) recognized by the cyclin subunit (as LxF motifs)(Loog & Morgan, 2005), and via the Cdk-binding subunits Cks1 or Cks2, which interact with phosphorylated threonine residues in primed substrates (Örd et al, 2019). These mechanisms promote processive multisite phosphorylation and allow Cdk1 to target substrates even at non-canonical sites. Our motif-based analysis captures only part of this complexity and may underestimate the number of true Cdk1 targets.

To further explore kinase involvement across phosphosite classes, we extended the analysis to all clusters and identified the most enriched kinase predictions for each (lines 360-365, new Figure 3—figure supplement 1). In Class II, the most enriched kinases included Cdk1, Erk2, and Plk1, supporting the conclusions derived from the identification of the phosphosites of this Class. But others such as Cdk2, Cdk3, Cdk5, Cdk16, KIS, JNK1, and JNK3 were also identified.

(4) Figure 3B: Why do the authors show this kind of Table only for Class I, not Classes II-V? It would be informative to show candidate proteins in other classes.

We chose to present the candidate proteins from Class I in a table format because the number of phosphosites (136) was too small to allow a meaningful Gene Ontology (GO) enrichment analysis. Therefore, we manually curated the data and highlighted proteins whose Class I phosphosites are associated with specific biological processes. For Classes II–V, the higher number of phosphosites allowed us to perform GO enrichment analyses. Since several of the enriched processes were shared across different classes, and some proteins have phosphosites in multiple classes, we opted to organize the results by biological processes rather than by class. We agree with the reviewer that it is indeed valuable to highlight interesting proteins with Class II–V phosphosites. We have done so in Figures 4 through 8, using graphical representations instead of tables, in order to make the data more accessible and avoid long tables. Additionally, the Supplementary Figures provide detailed phosphorylation trends for many of the proteins discussed in the main figures.

(5) It would be nice if the authors compare this phospho-proteome in *Xenopus* oocyte maturation with that in mouse oocyte maturation (Sun et al. 2024) in terms of evolutional conservation of the phospho-proteomes.

We thank the reviewer for this suggestion. As now detailed in the manuscript, we compared our *Xenopus* phosphoproteome with the dataset from Sun et al. (2024) on mouse oocyte maturation using the the phospho_occupancy_matlab package, available as part of our code repository (https://github.com/elizabeth-van-itallie) previously described in (Van Itallie et al, 2025). We identified 408 conserved phosphorylation sites corresponding to 320 *Xenopus* and 277 mouse proteins (see new paragraph: lines 824-860, new Figure 9, Methods: lines 1011-1032 and 1060-1065, and Appendix 7). Phosphorylation dynamics across meiosis were significantly correlated between the species (Pearson r = 0.39, p < 0.0001; new Figure 9A), highlighting evolutionary conservation of the phosphoproteomes. Key phosphorylation events such as Plk1 at T201, Gwl at S467, and Erk2 at T188 increased in both species, consistent with activation of the Cdk1 and MAPK pathways (Figure 6B, new Figure 9A–B). We validated experimentally several of these phosphorylation sites by western blot (Erk2, Plk1, Fak1, and Akts1) and demonstrated their dependency on Cdk1 activation (new Figure 9C). Together, these new findings support the conservation of key phospho-regulatory mechanisms across vertebrate oocyte maturation.

Minor points:

(1) Reference lists: Please add Sun et al (2024) shown in line 115.

This important reference has been added (lines 115, 134, 313 and 826).

(2) Figure 1, red arrows for the inhibition: This should be ‘T’ shape for a better understanding of these complicated pathways.

We agree with the reviewer’s remark, and we have modified Figure 1.

(3) Line 236-238: The authors referred to the absence of Cdc6 in oocyte maturation in *Xenopus*. However, Figure 2C shows that Cdc6 belongs to a list of accumulating proteins with Orc1 and Ocr2 etc. and the authors did not discuss this discrepancy in the text. Please clarity the claim.

We apologize for the unclear wording in our text. The section of the manuscript regarding the pre-RC components may have been misleading. The text has been revised to clarify that Cdc6 was not detected in prophase-arrested oocytes by western blot and that it accumulates during meiotic maturation after MI, enabling oocytes to replicate DNA (lines 243-250).

(4) Line 306: Please add the link to phosphosite.org.

The link has been added (line 319).